# Most long-lived contrails form within cirrus clouds with uncertain climate impact

Andreas Petzold [1,2] ✉, Neelam F. Khan [1,3], Yun Li [1], Peter Spichtinger [4], Susanne Rohs [1], Susanne Crewell [3], Andreas Wahner [1] & Martina Krämer [4,5]

Contrail-cirrus is considered the most important component of aviation-induced climate impact. However, a reliable assessment requires better understanding of their radiative effects. Analysis of seven years of humidity observations by instrumented passenger aircraft shows that conditions promoting long-lived contrails are fulfilled most often in regions already covered by subvisible or visible cirrus: ~90% over the Northern midlatitudes and almost 100% in the Southeast Asian subtropics, approximately equally distributed among visible and subvisible cirrus clouds. A conceptual analysis shows that subvisible cirrus and clear-sky cover ~10% of the cruise altitude over Northern midlatitudes ( < 2% in the subtropics) and contrails within these regions are expected to cause additional warming. However, most contrails in the thicker, visible cirrus, only slightly enhance the cirrus warming effect or possibly reverse it to cooling. Our results suggest that potential flight rerouting concepts for contrail avoidance need to consider cirrus cloud coverage in addition to ice-supersaturation.

According to most recent analyses, global aviation operations contribute about 3.5% to the total net anthropogenic warming, quantified by the metric 'effective radiative forcing (ERF) of climate' which measures the way these effects perturb the earth-atmosphere energy balance relative to pre-industrialization[1]. The climate impact of aviation results from both the direct emissions of carbon dioxide ($CO_2$), water vapour ($H_2O$), nitrogen oxides ($NO_x$) and aerosol particles (soot, organics, and sulphate)[2], and from indirect effects like aviation-$NO_x$ induced ozone formation, aerosol–cloud interactions, and the formation of persistent contrails and contrail-cirrus,. With an ERF of 60 mW m$^{-2}$, contrail-cirrus is estimated to be the largest non-$CO_2$ ERF contributing 60% (present-day) to the aviation net total ERF of 100 mW m$^{-2}$ [3,4]. Other than the well-understood $CO_2$ effect and its small uncertainty, the quantified non-$CO_2$ effects of aviation are associated with large uncertainties, mostly for contrail-cirrus with a substantial associated uncertainty of ~70%[5]. In particular, the feedbacks of

contrail-cirrus radiative effects and surface temperature are largely unknown, but a most-recent model study including these effects reports that the climate sensitivity is lower for contrail-cirrus than for aviation-related $CO_2$ emissions[6].

Uncertainties of the climate effects of contrail-cirrus are largely related to unanswered questions about the extent to which contrails form inside cirrus clouds, their influence on cirrus cloud coverage and optical depth, and the radiative implications[7,8]. However, it is obvious that contrail-cirrus formed in clear sky or in or above pre-existing subvisible cirrus clouds have a stronger radiative effect than contrail-cirrus embedded in pre-existing visible cirrus, because the difference in the optical contrast with and without contrail is much larger. The related microphysical and optical properties of the contrail ice particles develop more complexly inside pre-existing cirrus clouds than in an otherwise clear sky[9–11], although contrail formation itself is not affected by pre-existing cirrus clouds[12].

[1]Institute of Climate and Energy Systems—Troposphere (ICE-3), Forschungszentrum Jülich GmbH, Jülich, Germany. [2]Institute for Atmospheric and Environmental Research, University of Wuppertal, Wuppertal, Germany. [3]Institute for Geophysics and Meteorology, University of Cologne, Cologne, Germany. [4]Institute for Atmospheric Physics, Johannes Gutenberg University Mainz, Mainz, Germany. [5]Institute of Climate and Energy Systems—Stratosphere (ICE-4), Forschungszentrum Jülich GmbH, Jülich, Germany. ✉e-mail: a.petzold@fz-juelich.de

The analysis of linear contrail occurrence over the Northern hemisphere in MODIS data for the year 2006 revealed that contrails were detected within an environment with ice clouds in 33% of the cases, compared to 46% with low liquid water clouds, and 17% in clear skies[13]. The same data set[14] yields that the net contrail radiative forcing (RF), i.e., the sum of short-wave and long-wave RF, during daytime (~13:30 LT) is 3.9 Wm$^{-2}$ for clear sky and 1.3 Wm$^{-2}$ for contrails in a column already filled with visible cirrus clouds. During night, when no shortwave RF occurs, the longwave RF for clear and cloudy scenes is 16.9 and 11.0 Wm$^{-2}$, respectively. Thus, daytime net contrail RF is increased by a factor of 3 while nighttime longwave RF still is increased by a factor of 1.5 for contrails formed in clear sky or subvisible cirrus, compared to those having formed inside, above or below pre-existing cirrus.

Two studies using CALIPSO observations and flight track recordings reported an increase in cloud optical depth by 22% after an aircraft had crossed the cirrus[15], and a statistically significant increase by as large as a factor of 2 in the concentration of ice crystals in clouds affected by aviation[16]. A recent modelling study on the impact of existing uncertainty in contrail cirrus optical depth on the resulting contrail-cirrus ERF reports an eightfold uncertainty[5].

Observation-based knowledge on long-lived contrails within cirrus clouds and their radiative effects is severely limited. To narrow that knowledge gap, we study the occurrence of conditions favouring the formation and existence of long-lived contrails inside pre-existing cirrus clouds from observations of relative humidity with respect to ice (RH$_{ice}$) taken on board of passenger aircraft equipped with suitable instrumentation. The formation of contrails occurs when the Schmidt-Appleman criterion (SAC)[17] is fulfilled, i.e. when the hot, humid exhaust emitted from an aircraft engine mixes rapidly with cold and humid ambient air, causing the relative humidity in the cooling plume to exceed saturation with respect to liquid water (RH$_w$ >100%). Only when liquid water saturation is reached at a certain point in time of the plume evolution, water droplets form on the available aerosol particles, which subsequently freeze to contrail ice particles during the strong cooling caused by further mixing with the surrounding air. The validity of SAC for determining the contrail formation threshold temperature was confirmed by extensive observational data[18].

After reaching the ambient temperature, the contrail ice particles grow or shrink in size depending on the ambient humidity. If the environment remains ice-supersaturated (RH$_{ice}$ ≥ 100%), the contrails can persist with lifetimes from 4 to >10 h[17,19]. If the ambient air is ice-subsaturated (RH$_{ice}$ <100%), the ice particles sublimate and the contrail dissolves. For a contrail to survive more than approx. 10 minutes and to spread out to radiatively impactful contrail-cirrus, ambient RH$_{ice}$ must be close to or above ice-saturation[7,17,20–22].

From an observation-based climatology of ice crystal number concentrations as well as from a model analysis of contrail formation conditions we know that long-lived contrails and contrail-cirrus exist at temperatures from 205 K to 230 K for ice-supersaturated (RH$_{ice}$ ≥ 100%)[23,24] and slightly ice-subsaturated (90% ≤ RH$_{ice}$ <100%) conditions[25,26]. Aircraft flying through an ice-supersaturated or even slightly subsaturated air mass will thus form a persistent contrail and finally a contrail-cirrus as soon as ambient air temperature is below 230–225 K, which is almost always the case for Northern midlatitude cruise altitudes. Only for warmer temperatures above 230 K, as met, e.g., in the subtropics or at lower altitudes in the midlatitudes, these regions may be crossed by commercial aircraft without generating a contrail.

Today, the occurrence of contrail-cirrus is discussed in relation to cold (T < 230 K) ice-supersaturated regions (ISSR: RH$_{ice}$ ≥ 100%). To include the observationally confirmed existence of contrail-cirrus also at slight ice-subsaturation in the consideration of aviation climate impacts, we introduce the term "potential contrail-cirrus region" (PCCR) which we define as an air mass with RH$_{ice}$ ≥ 90% and SAC

fulfilled. In such an air mass, a contrail would form and develop through the stage of ice-supersaturation during continuing mixing of hot exhaust with ambient air, where the ice crystals grow. The resulting contrail, formed of larger ice crystals, would then transition to sublimation with a limited but long-enough lifetime for becoming radiatively impactful. Model calculations of ice crystal sublimation in a dissolving contrail demonstrated that it can take approximately 4 h for ice particles to sublimate until RH$_{ice}$ declines to below 80%[26]. Therefore, the "embedded contrails" is used synonymously with PCCR.

Thus, PCCRs include ISSRs and represent areas with the potential for contrail and contrail-cirrus of lifetimes ranging from 4 to over 10 h when crossed by an aircraft. A first observation-based estimate of the areal fraction of air masses prone to contrail-cirrus formation for the North Atlantic flight corridor yielded an increase from 30% for ISSRs to 43% when considering RH$_{ice}$ ≥90% as a sufficient pre-requisite for contrail-cirrus existence, i.e., for PCCRs[26]. This significant increase in the fractional area justifies the definition of a specific term to allow the differentiation from ISSRs.

The formation conditions and occurrence characteristics of ice-supersaturated regions (ISSRs) as well as their links to the formation of contrail-cirrus have been intensely studied for both clear-sky and in-cloud conditions[27–31]. Vertically, ISSRs occur most frequently in the upper troposphere just below the tropopause layer, whereas the upper free troposphere farther below the tropopause layer becomes less humid with respect to ice. However, the prediction of ISSRs is difficult because they are highly variable in space and time and depend crucially on the driving weather systems[32–34]. As a further complication, forecast models suffer from an underrepresentation of ice-supersaturation in the upper troposphere[31,35] which is compensated by assuming model fields of RH$_{ice}$ > 93% as ice-supersaturated[36] or adjust humidity model fields to observations[37,38]. In addition, a not yet achieved precision of representation of ISSRs in numerical weather models[9,31,36,39] is needed for a better forecast of ISSRs. The known difficulties in predicting ISSRs are most certainly valid also for PCCRs. It also needs to be mentioned that forecasting cirrus cloud formation and its properties is challenging for the same reasons as those for ISSRs and potentially PCCRs, since ISSRs are an indispensable pre-requisite for cirrus cloud formation.

There are several measures discussed for reducing aviation's climate impact, including technological advancements for reducing the emissions of short-lived climate forcers such as NO$_x$ and aerosols, and the availability of sustainable aviation fuels for reducing long-lived greenhouse gases[40]. On the short term, operational concepts for avoiding contrails have been proposed as a strategy for reducing aviation's climate impact[41,42]. However, they are intensively discussed with respect to potential trade-offs between longer flight trajectories for avoiding potential contrail-cirrus regions and resulting additional CO$_2$ emission from extra fuel burn[43,44]. It is also not clear how atmospheric conditions in future climate may have an impact on the occurrence of PCCRs[45]. Additionally, the latest IPCC report highlights the role of clouds in a warming climate[46] which makes the reduction of anthropogenically generated high ice clouds even more urgent.

Here, we provide an extensive analysis of contrail-cirrus occurrence in clear sky, as well as in subvisible and visible cirrus for the two densest air spaces globally and discuss the consequences of moving from ISSRs to PCCRs for the fractional air masses prone to a strong aviation climate impact. Additionally, we discuss the potential radiative effects of contrails in clear sky as well as in the presence of subvisible, visible and optically thick cirrus clouds. The data base of this study builds on about 7 years of continuous in-situ observations of temperature and RH$_{ice}$ by the European research infrastructure IAGOS[47] which measures these properties along with atmospheric composition by passenger aircraft carrying scientific instrumentation during their regular operations.

## Results

The data analysis strategy behind our results relies on four pillars: (1) the distribution of $RH_{ice}$ from in-situ observations, and the subsequent determination of areal fractions, or occurrence probabilities, respectively, fulfilling PCCR or ISSR conditions; (2) fulfilment of SAC using in-situ measured temperature data for the identification of regions where contrails will form; (3) cirrus cloud coverage from ERA5 reanalysis cloud ice water content (CIWC), which can reliably capture cloud presence by the assimilation of comprehensive satellite radiance measurements; and (4) the distinction between areas covered by clear sky or subvisible clouds and areas covered by visible clouds, by means of a CIWC-based threshold value separating subvisible from visible cirrus.

The expected radiative effects in terms of warming or cooling by contrail-cirrus in clear sky or embedded in existing cirrus clouds are illustrated in Fig. 1 which has been adapted from Krämer et al.[23]. The lower and upper bounds of the cooling or warming effects of cirrus clouds as a function of cloud optical depth result from radiative transfer calculations. Since only a single set of conditions, (noon at solar zenith angle at 50° North) was used and the results for other angular conditions, latitudes, and backgrounds will be highly variable, the figure only serves as a descriptive sketch. The radiative effects of contrail-cirrus developing in the various environments are categorised in the context of radiative effects of the embedded contrail-cirrus in addition to the radiative effects of the unperturbed clouds.

The qualitative categorisation of the climate impact of contrail-cirrus for the various environments considers five contrail development scenarios: if only short-lived contrails with low cloud optical depth (OD) form, there is no resulting climate impact (Case 0); contrails formed in clear sky developing into visible contrail-cirrus lead to net warming (Case 1); contrails formed in subvisible cirrus developing into visible contrail-cirrus lead to stronger net warming (Case 2); contrails formed within visible cirrus clouds (embedded contrail-cirrus) only lead to an additional minor but still net warming effect (Case 3); contrails forming in already optically thick cloud can enhance optical depth to overall cooling effect (Case 4). The net warming of Cases 1 and 2 and the minor net warming of Case 3 are confirmed by MODIS observations[14], whereas a model study[11] speculates that ignoring the impact of cirrus on contrail formation may possibly

underestimate the cooling of the contrail perturbations within optically thicker clouds (Case 4).

In summary, we conclude that areas covered by clear sky (Case 1) or subvisible cirrus (Case 2), with SAC fulfilled, and at least qualified as PCCR with $RH_{ice} > 90\%$ are associated with enhanced positive contrail RF causing stronger warming (category "warming). Areas already covered by visible cirrus clouds (Case 3, Case 4) are associated with an ambiguous RF effect, ranging from minor warming to cooling (category "ambiguous"). These difference needs to be kept in mind in the following when we distinguish three different categories: clear sky, subvisible cirrus, and visible cirrus which are distinguished using thresholds in CIWC (see 'Methods').

### PCCRs and ISSRs in major air traffic regions

This IAGOS data set provides unprecedented in-situ measurements taken exactly in the regions of major air traffic. The fractions of all PCCRs and ISSRs were quantified for the densest airspace globally between North America and Europe (approx. 10 million IAGOS flight kilometres; see Fig. 2a and 'Methods') in the Northern midlatitudes and, for comparison, for the second densest airspace globally over Southeast Asia (approx. 6 million IAGOS flight kilometres) in the subtropics. For the Northern midlatitudes, we further specified the analysis for Eastern North America, North Atlantic and Western Europe to illustrate the variability of our findings for areas over land and over the ocean; see Figure S1 in the supplementary information.

The vertical distribution of data used in our study is illustrated in Fig. 2b relative to the pressure of the thermal tropopause according to WMO[48] ($p_{TTP}$) which separates the dry lowermost stratosphere with an almost negligible potential of contrail formation from the humid and cold tropopause layer and uppermost troposphere. The definition of the pressure levels follows established IAGOS concepts[30,49] and is described in 'Methods'. The resulting vertical distributions of occurrence of PCCRs and ISSRs are illustrated in Fig. 2c, d as fractional occurrences per pressure level. Obviously, the occurrence frequency of PCCRs−and associated long-lived contrails−is by about 10% higher than that of ISSRs.

While the cruising levels of civil aviation fall within a certain pressure range globally, the thermal tropopause in the subtropics is located at higher altitudes than in the midlatitudes. Consequently, in the extratropics, e.g., at Northern midlatitudes, civil aviation cruising altitude overlaps with the cold and humid layers just below the thermal tropopause, where cirrus clouds are frequent[28,30], while in the subtropics, the main cruising levels are located deeper inside the troposphere where the temperatures are warmer and SAC is not fulfilled. Thus, the occurrence probability of PCCRs and ISSRs is reduced by a factor of three or larger.

### Distribution of potential contrail-cirrus in clear sky and inside natural cirrus clouds

The differences in the vertical distributions of cruising levels and atmospheric layers of frequent PCCRs and ISSRs occurrence between midlatitudes and subtropics have immediate consequences on all aviation non-$CO_2$ effects and particularly for contrail-cirrus. This includes both clear-sky and cloudy conditions which are distinguished by a CIWC threshold of $CIWC_{clear} = 0.001$ ppmv, marking the onset of subvisible cirrus (see 'Methods'). Conditions favouring contrail-cirrus existence at cruise altitude are met frequently in the midlatitudes, while they are sparsely met in the subtropics, which becomes clearly visible in Fig. 3. For the midlatitudes (Fig. 3a), the fraction of air masses in which no contrails can form (SAC not fulfilled), is of the order of 1% of the observations made, i.e., indistinguishable from the data noise level, while for the warmer subtropical region they can reach up to 15% for both clear-sky and in-cloud conditions.

Conditions favouring the formation of short-lived contrails (defined as SAC fulfilled, $RH_{ice} < 90\%$) dominate in all regions with

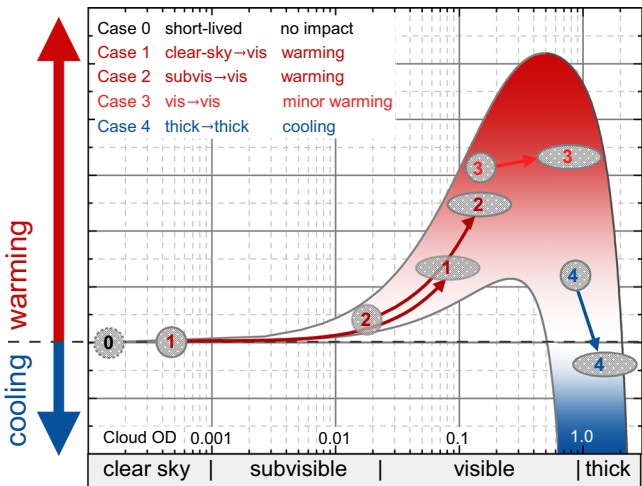

**Fig. 1 | Schematic of the radiative impact of contrail-cirrus embedded in pre-existing cirrus.** Contrail formation scenarios are: Case 0−formation of short-lived contrails; Case 1−contrail formed in clear sky develop into visible contrail-cirrus; Case 2−contrail formed in subvisible cirrus develops into visible contrail-cirrus; Case 3−contrails formed within visible cirrus clouds develops into embedded contrail-cirrus; Case 4: contrail formed in optically thick cloud can further enhance cirrus cloud optical depth (cloud OD).

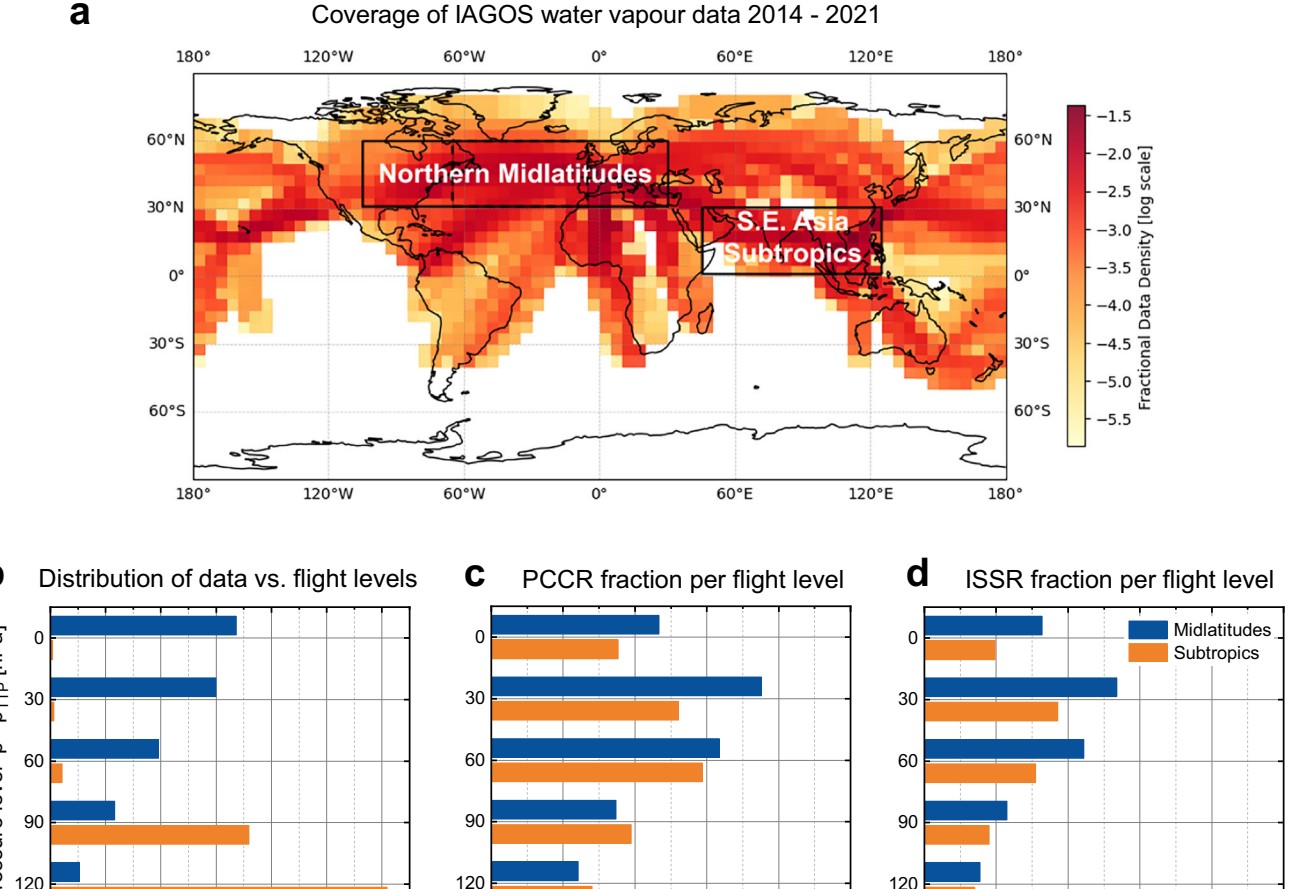

**Fig. 2 | Global coverage of IAGOS water vapour data for the period 2014 – 2021.**
**a**: Fractional data density of IAGOS combined water vapour dataset from June 2014 to December 2021 (logarithmic scale used to visualize maximum data coverage); inserted boxes indicate the regions of interest; **b**: vertical distribution of data for five pressure levels of thickness 30 hPa relative to the pressure level of the thermal tropopause $p_{TTP}$; **c** fraction of potential contrail-cirrus regions (PCCR; relative humidity with respect to ice $RH_{ice} \geq 90\%$) for each pressure level; **d** same as **c** but for ice-supersaturated regions (ISSR; $RH_{ice} \geq 100\%$).

fractions of 70% and more of the probed air masses (sum of dark and light blue bars in Fig. 3), indicated as short-lived. Long-lived contrails, i.e., those fulfilling the SAC criterium and exceeding the $RH_{ice}$ threshold naturally occur more frequently for PCCR than for ISSR. For the midlatitudes and for both threshold $RH_{ice}$ values, however, the number of persistent contrails or contrail-cirrus forming inside pre-existing clouds (light blue bars in Fig. 3, indicated as long-lived) exceeds the number of cases in clear sky (respective dark blue bars in Fig. 3) by at least a factor of five, whereas in the subtropics, contrail persistence is only met inside clouds.

The total fraction of long-lived contrails in PCCRs (ISSRs) for both clear-sky and in-cloud conditions with respect to all probed air masses, is on average 28% (20%) for the Northern midlatitudes (Fig. 3a) and varies between 23% (15%) over Western Europe and 33% (24%) over the North Atlantic (Fig. 3c–e). Our observations over Europe are in good agreement with results from radiosonde-based studies[50,51], while for the North Atlantic no reference data are available. Overall, the increase in the fractional area prone to contrail-cirrus formation is almost 10% for the midlatitudes when considering PCCR instead of ISSR only, which is in accordance with earlier analyses of IAGOS data[26]. For Southeast Asia (Fig. 3b), the respective PCCR (ISSR) fraction is less than 14% (7%), reflecting the larger distance and warmer temperatures of cruising altitudes to the thermal tropopause and thus reduced $RH_{ice}$ levels compared to the midlatitudes.

As discussed in the context of Fig. 1, the climate impact of contrail-cirrus depends crucially on the cloudiness of its environment. For the transition from subvisible to visible cirrus lidar observations[52] define an optical thickness of 0.03 while theoretical calculations yield a value between 0.01 and 0.015 for a 500 m thick cloud[53]. Here, cloud optical depth is derived from the cloud ice water content through the relationship between cloud optical depth, CIWC and ice crystal effective diameter ($D_{eff}$) for different CIWC and $D_{eff}$ values; details are described in 'Methods'. We applied the CIWC threshold values $CIWC_{vis}$ for cloud visibility of 1.0 ppmv and 2.0 ppmv to quantify the dependence of our results on the doubling of the CIWC visibility threshold value. A statistical analysis of the conditions and the connected climate impact of these three formation categories (clear sky, subvisible, visible) is given in Table 1. Details on the definition of the regions, CIWC categorization, and the results of the full statistical analysis are given in 'Methods' and in Table S1 of the supplementary information.

From this categorization it can be deduced that in the Northern midlatitudes, PCCR (ISSR), i.e., conditions prone to contrail-cirrus formation are found for max. ~4% (~2% or less) of the studied area in clear sky, ~9% (~6%) inside subvisible cirrus clouds, and ~15% (~12%) inside visible cirrus clouds (Fig. 3a). For the subtropics, the distribution is even more pronounced with almost all PCCR and ISSR conditions observed inside visible clouds, while observations in clear sky or

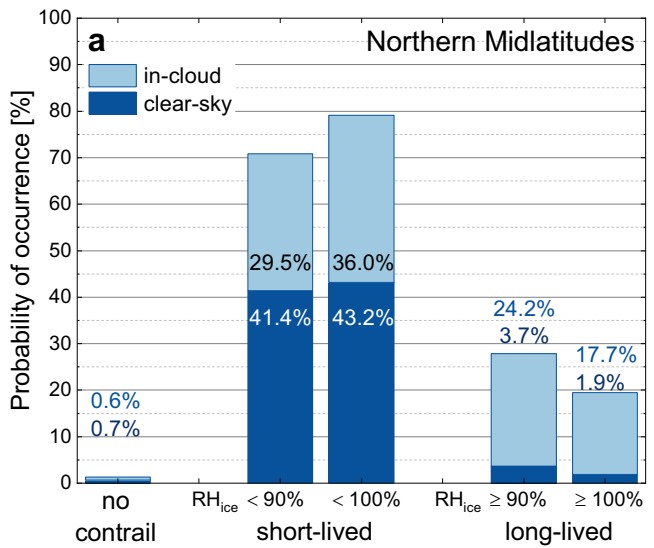
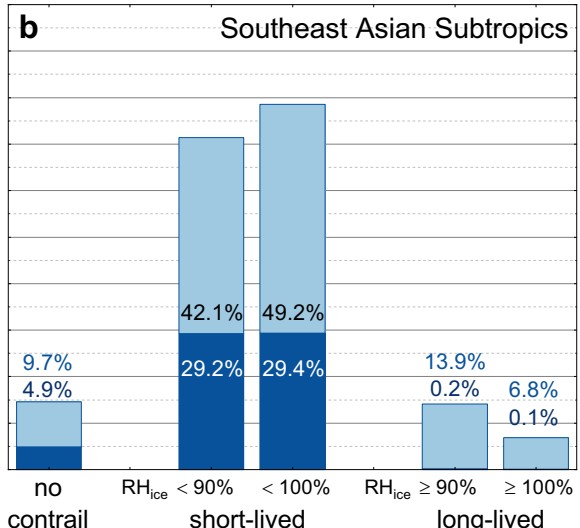

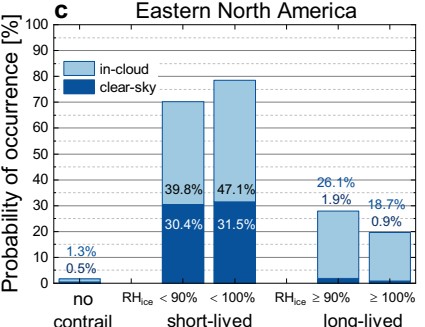
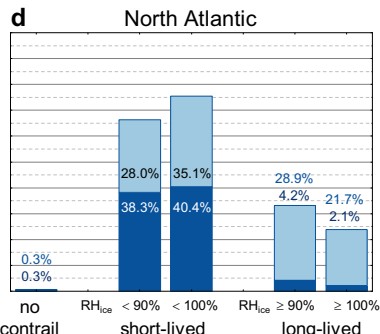
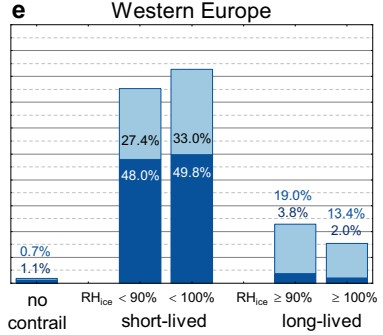

**Fig. 3 | Occurrence probability of contrail formation conditions in clear sky and inside clouds.** Fraction of air masses for the regions Northern Midlatitudes (**a**), Southeast Asian Subtropics (**b**), Eastern North America (**c**), North Atlantic (**d**), and Western Europe (**e**), promoting no contrail formation (Schmidt-Appleman criterion not fulfilled), formation of short-lived contrails (Schmidt-Appleman criterion fulfilled, relative humidity with respect to ice $RH_{ice}$ < threshold $RH_{ice}$) and formation of persistent contrails and contrail-cirrus (Schmidt-Appleman criterion fulfilled, $RH_{ice} \geq$ threshold $RH_{ice}$, 90% for PCCRs, 100% for ISSRs); dark blue bars indicate clear-sky conditions (cloud ice water content CIWC < $CIWC_{clear}$ = 0.001 ppmv), light blue bars in-cloud conditions (CIWC $\geq CIWC_{clear}$).

subvisible clouds contribute max. 1% which, however, is considered below the noise level (Fig. 3b).

Recalling that the largest part of the studied regions is covered by air masses well below ice-saturation, conditions promoting long-lived contrails (PCCR, ISSR) are fulfilled most often in regions already covered by cirrus. If all air masses at or above the clear-sky threshold ($CIWC \geq CIWC_{clear}$) are considered (Fig. 3a, category long-lived), then between 87% (PCCR conditions) and 90% (ISSR conditions) of contrails form either in sub-visible or visible clouds. If only the visible clouds with $CIWC_{vis} \geq 2.0$ ppmv are taken into account, then these fractions reduce to 46% and 52%; see the right column in Table 1. In the Southeast Asian subtropics, contrail-cirrus conditions are met only inside clouds (Fig. 3b, category long-lived). See Table 1 for details and Fig. 3c–e for the variability across the Northern midlatitudes. Numerical values are provided in Table S2 of the supplementary information.

The above-mentioned study on properties of linear contrails using MODIS data[13] reports 33% of contrails formed in areas already covered by clouds. The applied method distinguishes contrails from other cirrus by their linear shape and relatively large 11–12 μm brightness temperature difference. We apply a combination of in-situ based criteria such as the fulfilment of SAC, exceedance of $RH_{ice}$ threshold for PCCR, and exceedance of ERA5 $CIWC_{vis}$ to an on-flight-level dataset and obtain a fraction of 46% (PCCR, $CIWC_{vis} = 2.0$ ppmv). Taking into account that about ~20% of clouds classified in the MODIS analysis as

water were likely to consist of non-opaque cirrus over lower liquid water clouds and a few more water clouds were misclassified as ice than ice clouds misclassified as water[54], the agreement between the two studies is reasonable.

## $RH_{ice}$ distributions in clear sky and inside cirrus clouds

The distribution of $RH_{ice}$ inside clouds is important for the understanding of the respective cloud types. Figure 4a illustrates the frequencies of measured $RH_{ice}$ for the midlatitude regions with their peak $RH_{ice}$ values slightly above 100% inside visible cirrus clouds (blue shaded area), mostly below 100% in subvisible cirrus (light-blue shaded area), and predominantly at low $RH_{ice}$ values in clear sky (red line). Figures 4c–e show the variability of $RH_{ice}$ distributions for the different study regions of the Northern midlatitudes. While the overall features maintain, details of the split between subvisible and visible clouds vary between over-land and over-ocean regions. We interpret our observations of the larger fraction of subvisible cirrus in subsaturation such that these cirrus clouds are mostly slowly dissolving clouds, while visible cirrus clouds are more often in the developing stage at or above ice-saturation.

For the Southeast Asian Subtropics (Fig. 4b), the peak value of $RH_{ice}$ inside clouds is found at slightly sub-saturated conditions, while subvisible clouds are almost not present. That can be understood because (i) in the subtropics the cruising level of civil aviation is farther below the

**Table 1 | Regional split of coverage of areas with Schmidt-Appleman criterion (SAC) fulfilled and further subdivided into classes of potential climate impact for areas with SAC fulfilled, and respective contrail-cirrus overlap for both potential contrail-cirrus regions (PCCR) and ice-supersaturated regions (ISSR); data are based on IAGOS in-situ observations of relative humidity with respect to ice ($RH_{ice}$) and temperature, SAC was calculated from IAGOS temperature observations, and ERA5[60] cloud ice water content (CIWC) provided the cloud categorisation**

| Region | Threshold $RH_{ice}$ | clear sky CIWC < 0.001 ppmv | subvisible cirrus 0.001 ppmv ≤ CIWC < 1.0 ppmv (2.0 ppmv) | visible cirrus CIWC ≥ 1.0 ppmv (2.0 ppmv) | Climate impact total | Climate impact warming | Climate impact ambiguous | Climate impact no | Contrail – cirrus overlap CIWC visibility threshold 1.0 ppmv | 2.0 ppmv |
|---|---|---|---|---|---|---|---|---|---|---|
| Column | | a | b | c | a + b + c | a + b | c | 100 – (a + b + c) | 100 c / (a + b + c) | |
| Northern Midlatitudes | PCCR (≥ 90%) | 3.7% | 9.0% (11.5%) | 15.2% (12.8%) | 28.0% | 12.7% (15.1%) | 15.2% (12.8%) | 72.1% | 54.5% | 45.7% |
| | ISSR (≥100%) | 1.9% | 5.9% (7.6%) | 11.8% (10.1%) | 19.6% | 7.8% (9.5%) | 11.8% (10.1%) | 80.4% | 60.2% | 51.5% |
| SE Asian Subtropics | PCCR | 0.2% | 1.1% (1.6%) | 12.7% (12.2%) | 14.0% | 1.4% (1.9%) | 12.7% (12.2%) | 85.9% | 90.7% | 87.1% |
| | ISSR | 0.1% | 0.5% (0.8%) | 6.3% (6.0%) | 6.9% | 0.6% (0.9%) | 6.3% (6.0%) | 93.1% | 91.3% | 87.0% |

ice-supersaturated layer near the tropopause where cirrus form regularly, and (ii) the relative occurrence of ice-subsaturated air masses inside clouds increases with the distance from the cloud top[55]. Thus, in the subtropics, contrail-cirrus occur almost exclusively in physically and optically thick clouds closer to the cloud base. For clear-sky conditions, the most frequently observed $RH_{ice}$ values represent dry conditions with $RH_{ice} < 25\%$ at temperatures above 225 K, while the coldest air masses probed at temperatures below 215 K are frequently close to ice-saturation; see Fig. S2 and discussion in the supplementary information.

$RH_{ice}$ distribution patterns like the IAGOS-based observations have been reported for cirrus clouds in the midlatitude uppermost troposphere[20,49,56], but without the separation of subvisible from visible cirrus. The cloud categorisation successfully applied here, in combination with the visibility criterion now allows the separation of clear-sky and in-cloud conditions with consideration of cloud thickness and thus optical depth.

## Discussion

The effects of contrails interacting with pre-existing cirrus clouds are generally known[22] but have not yet been considered in the quantification of the contrail-cirrus climate impact[6,8,9]. To start with, model studies reported that pre-existing cirrus have no impact on the formation of contrails, that contrail formation within cirrus mostly leads to increasing cirrus ice crystal numbers, that cirrus ice crystals mixed into the evolving contrail do not efficiently slow down the sublimation of the ice crystals in the downward travelling wing vortex, and that cirrus can dissolve an embedded contrail after few hours by aggregation of ice crystals[11,12]. The changes in the radiative properties of subvisible and visible cirrus resulting from the above-described changes of cloud microphysics after the release of a contrail of mean optical depth of 0.34 (median 0.24, modal 0.1)[57] into the cloud will increase the resulting optical depth of the modified cirrus (see Fig. 1), but with yet unknown radiative impact.

To quantify the potential climate impacts of contrail-cirrus in clear sky, subvisible or visible cirrus as shown in Fig. 3 and detailed in Table 1, we analysed the frequencies of areas with (i) no potential climate impact (SAC not fulfilled, or only short-lived contrails), (ii) ambiguous (minor net warming or cooling) climate impact (pre-existing visible cirrus: SAC fulfilled, $RH_{ice} >$ threshold $RH_{ice}$, and CIWC ≥ $CIWC_{vis}$), and (iii) net warming climate impact (clear-sky: SAC fulfilled, $RH_{ice} >$ threshold $RH_{ice}$, and CIWC < $CIWC_{clear}$; or subvisible cirrus: SAC fulfilled, $RH_{ice} >$ threshold $RH_{ice}$, and $CIWC_{clear} \leq$ CIWC < $CIWC_{vis}$). The results, summarised in Fig. 5 and presented numerically in Table 1, middle column, highlight the expected climate impact. This information supplements that given in Fig. 3 which focuses on the environmental conditions of contrail-cirrus occurrence. They clearly indicate that in the Northern midlatitudes, areas of net warming climate impact by contrail-cirrus (red bars) occur only in less than 15% ($CIWC_{vis}$ = 2.0 ppmv) of all observed cases for PCCR conditions and in less than 10% for ISSR conditions and are almost not observed over Southeast Asia. For a lower CIWC threshold for cloud visibility of 1.0 ppmv, the fraction of areas of net warming climate impact decreases slightly to less than 13% for PCCR and below 8% for ISSR conditions.

In-situ formed cirrus clouds in slow or fast updraft regions are mostly responsible for the optically thin or even subvisible warming cirrus clouds, whereas optically thick, mostly liquid-origin cirrus clouds have a potential cooling effect[23]. From a trajectory-based classification of ERA-Interim ice clouds in the region of the North Atlantic storm track[58] it is known that in-situ formed cirrus clouds with an associated stronger climate warming impact are more than twice as abundant as liquid-origin cirrus clouds above 300 hPa and reach near totality above 200 hPa, independent of the season. Thus, we conclude that at typical cruising altitude pressure levels between 200 and 245 hPa[26], areas of a stronger warming effect of contrail-cirrus can likely be associated with clear-sky and in-situ origin cirrus regions.

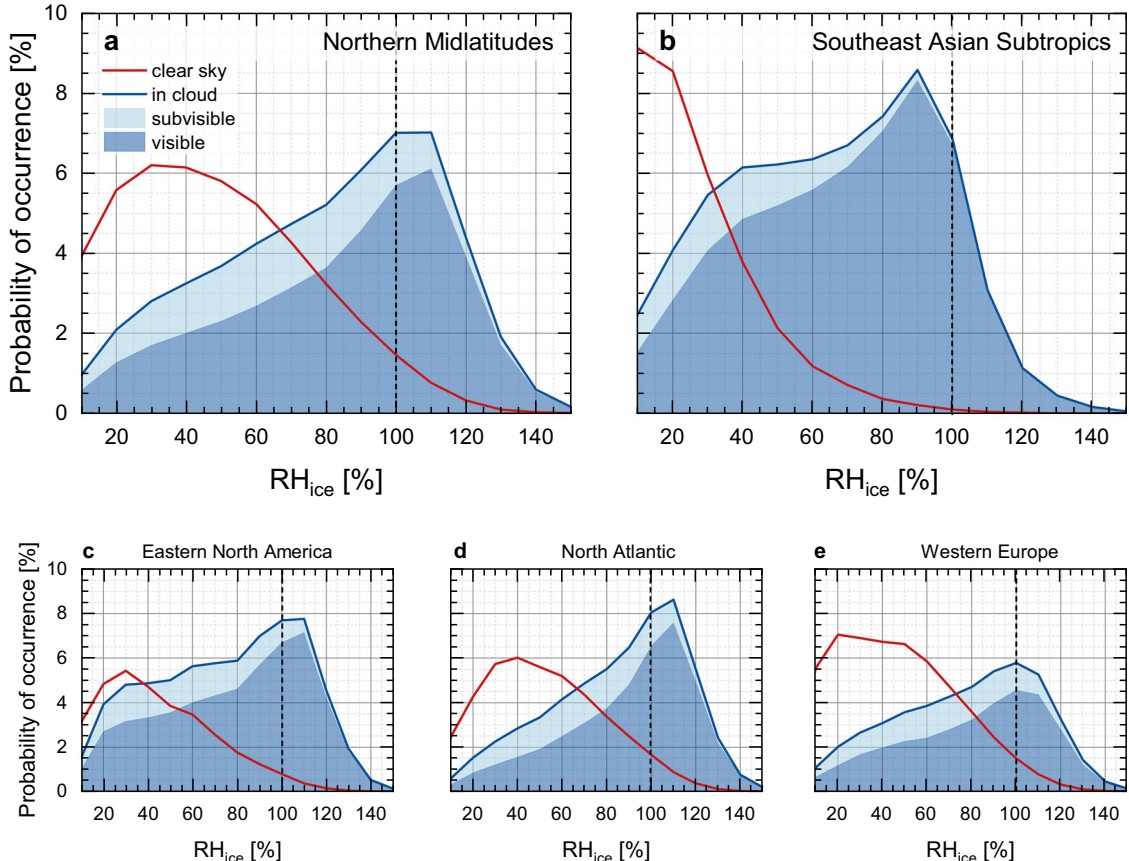

**Fig. 4 | Occurrence probabilities of RH$_{ice}$ values obtained from IAGOS in-situ observations.** The panels represent averaged probability distribution functions of relative humidity with respect to ice (RH$_{ice}$) form IAGOS observations below the thermal tropopause for the regions Northern Midlatitudes (**a**), Southeast Asian Subtropics (**b**), Eastern North America (**c**), North Atlantic (**d**), and Western Europe (**e**); red lines represent clear-sky conditions with cloud ice water content CIWC < 0.001 ppmv, dark blue lines in-cloud conditions for CIWC ≥ 0.001 ppmv; subvisible clouds refer to CIWC < 1.0 ppmv, visible clouds to CIWC ≥ 1.0 ppmv.

Finally, Fig. 6 displays the regional and seasonal distributions of PCCRs in the Northern midlatitudes with warming (panels a, c) and ambiguous (panels b, d) climate impacts, following the definitions above. In winter months, areas covered with visible and optically thick cirrus predominate independent of latitude, where embedded contrail cirrus have an ambiguous climate impact. In the summer months, these areas occur only at lower latitudes. For contrail-cirrus embedded in subvisible cirrus with stronger net warming impact the differences between seasons are less pronounced, but again with a slight tendency towards higher frequencies in winter months.

The results of our study suggest that conditions necessary for the formation of contrail-cirrus are most often fulfilled in regions that are already covered by visible cirrus, which reduces their warming impact or even turns it into a cooling effect. Those of a stronger warming significance, however, form under only limited circumstances in clear sky and subvisible cirrus. Unfortunately, our knowledge on the radiative effects of contrail-cirrus embedded in subvisible, visible and optically thick clouds, and feedback mechanisms between contrail and cirrus ice particles is highly limited. There will be a range of radiative forcings that could be produced by embedded contrail-cirrus, and the quantification of the climate impacts would require a sensitivity study and possibly consideration of low-level clouds.

This topic needs to be further analysed and verified in terms of their radiative impact for consideration of the potential impact of contrail avoidance strategies including flight rerouting. Finally, ice-supersaturation only is by far not a reliable, stand-alone criterion for assessing contrail-cirrus climate effects. Instead, we need to include

pre-existing cirrus clouds and their interactions with contrail-cirrus into all our considerations.

## Methods

### IAGOS in-situ RH$_{ice}$ and cloud datasets, ERA5 cloud ice water content

The analysed data set covers the period from June 2014 to December 2021 and contains in total more than $17 \times 10^6$ data points with each datapoint corresponding to 4 s sampling time, or 1 km flown distance at an average cruising speed of 250 m s$^{-1}$, respectively. Figure 2a shows the normalized data density per 5° x 5° grid. Four regions of interest were identified for in-depth statistical analyses, with three of them located in the Northern midlatitudes (30–60°N), namely Eastern North America (Region 1: 105–65°W, $1.57 \times 10^6$ data points), the North Atlantic flight corridor (Region 2: 65–5°W, $4.43 \times 10^6$ data points), and Western Europe (Region 3: 5°W–30°E, $4.72 \times 10^6$ data points), and one in the Southeast Asian subtropics (Region 4: 0–30°N, 45–120°E, $6.14 \times 10^6$ data points). Here, we mainly refer to the total of regions 1–3 as Northern midlatitudes but also show the results for the individual regions to illustrate the variability depending on geographical differences. The Southeast Asian tropical and subtropical region is of potential interest because of the high traffic density, though the cruise levels are farther below the tropopause level and thus at warmer temperatures than in the midlatitudes where cruise altitudes are close to the tropopause; see Fig. 2b.

The combined IAGOS RH$_{ice}$, air temperature, and ERA5 CIWC dataset spans from June 2014 to December 2021. To restrict the analysis on the air masses of the upper troposphere and the

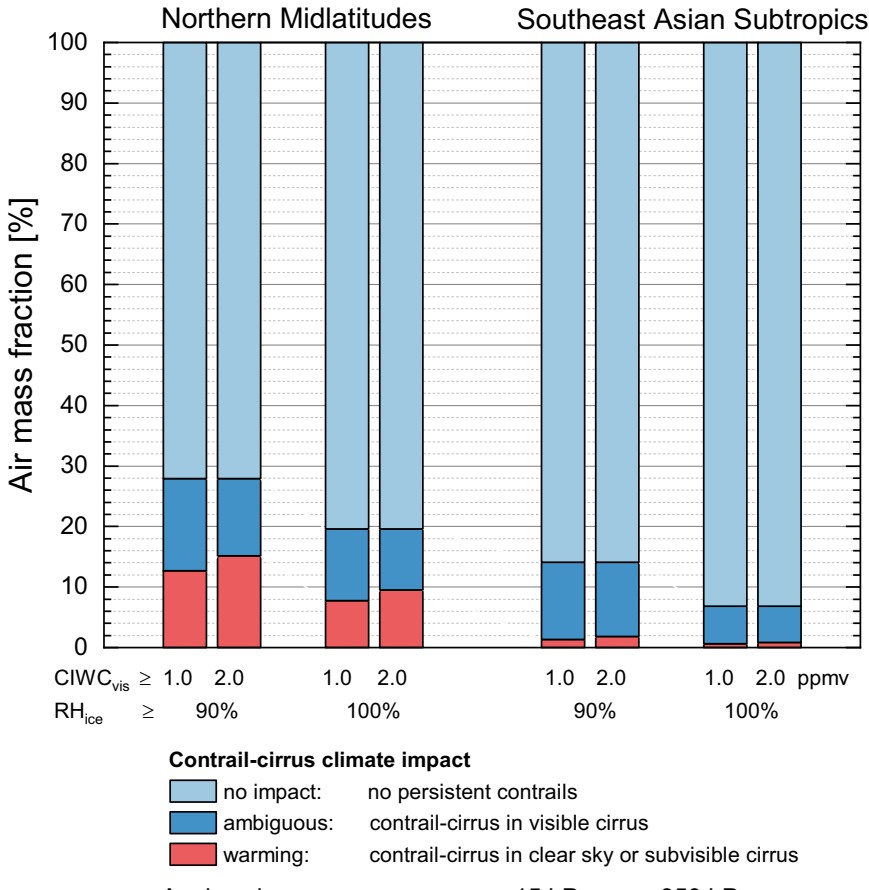

**Fig. 5 | Occurrence of air masses with different contrail-cirrus climate impacts.** Fractions of air masses with no aviation climate impact (Schmidt-Appleman criterion not fulfilled or short-lived contrails only), ambiguous climate impact (contrail-cirrus in visible cirrus with cloud ice water content CIWC ≥ 1.0 ppmv and ≥ 2.0 ppmv), and net warming climate impact (contrail-cirrus in clear sky or in subvisible cirrus). All analyses were performed for both potential contrail-cirrus regions (PCCR; relative humidity with respect to ice $RH_{ice} ≥ 90\%$) and ice-supersaturated regions (ISSR; $RH_{ice} ≥ 100\%$) and CIWC visibility threshold values ($CIWC_{vis}$) of 1.0 ppmv and 2.0 ppmv. Analysed pressure levels reached from 15 hPa below the pressure at the thermal tropopause ($p_{TTP}$) up to 350 hPa. Detailed numbers are listed in Table 1, middle column.

tropopause layer with respect to the thermal tropopause (TTP) and to focus on ice clouds only, data were selected based on the following criteria[30]:

1. Pressure below 350 hPa (~8.1 km) to focus on cruise altitude,
2. Pressure above $p_{TTP} − 15$ hPa to exclude dry stratospheric air masses that are too dry for cirrus clouds, and
3. Air temperature below 235 K (the threshold for spontaneous freezing of water droplets) to exclude air masses containing supercooled liquid water droplets.

According to WMO[48], the thermal tropopause (TTP) is defined as the lowest level at which the lapse rate decreases to 2 K km$^{-1}$ or less. In our study, vertical atmospheric layers relative to the thermal tropopause are defined as:

Thermal tropopause layer TTP : $p = p_{TTP} ± 15$hpa

Upper troposphere layer UT1 : $p_{TTP} + 15$hpa $< p < p_{TTP} + 45$hpa

Upper troposphere layer UT2 : $p_{TTP} + 45$hpa $< p < p_{TTP} + 75$hpa

Upper troposphere layer UT3 : $p_{TTP} + 75$hpa $< p < p_{TTP} + 105$hpa

The separation of UT layers was only used for the analysis of the PCCR/ISSR occurrence with altitude (Fig. 2b). For all other analyses, the pressure band reaching from 350 hPa down to $p_{TTP}$ -15 hPa was considered.

We quantified the contrail formation conditions for all sampled air masses by applying the Schmidt-Appleman criterion (SAC)[17], and added the information on cloudiness by applying the herein developed cloud categorisation based on ERA5 Cloud Ice Water Content (CIWC). This means that if a cloud is detected, there is also a contrail in the air mass, since the data points are based on real flight paths. The observations were then sub-divided into clear-sky and in-cloud sequences (clear sky: CIWC < 0.001 ppmv, subvisible cirrus: 0.001 ppmv ≤ CIWC < 1.0 ppmv (2 ppmv), visible cirrus: CIWC ≥ 1.0 ppmv (2 ppmv), see Table 1). For each subset, conditions were rated as "no contrail formation" if SAC was not fulfilled, as favouring the formation of "short-lived contrails" if SAC was fulfilled and $RH_{ice}$ <threshold $RH_{ice}$ (threshold for contrail-cirrus occurrence, either 90% or 100%, see Table 1), and as favouring the formation of "contrail-cirrus" if SAC was fulfilled and $RH_{ice}$ > threshold $RH_{ice}$. The complete overview over the statistical analysis is compiled in Table S1 of the supplementary information.

The in-situ $RH_{ice}$ data are calculated from direct measurements of relative humidity with respect to liquid water ($RH_{liq}$) and ambient temperature by the IAGOS Capacitive Hygrometer (ICH). The ICH combining a thin-film HUMICAP® capacitive sensor (Vaisala) with a

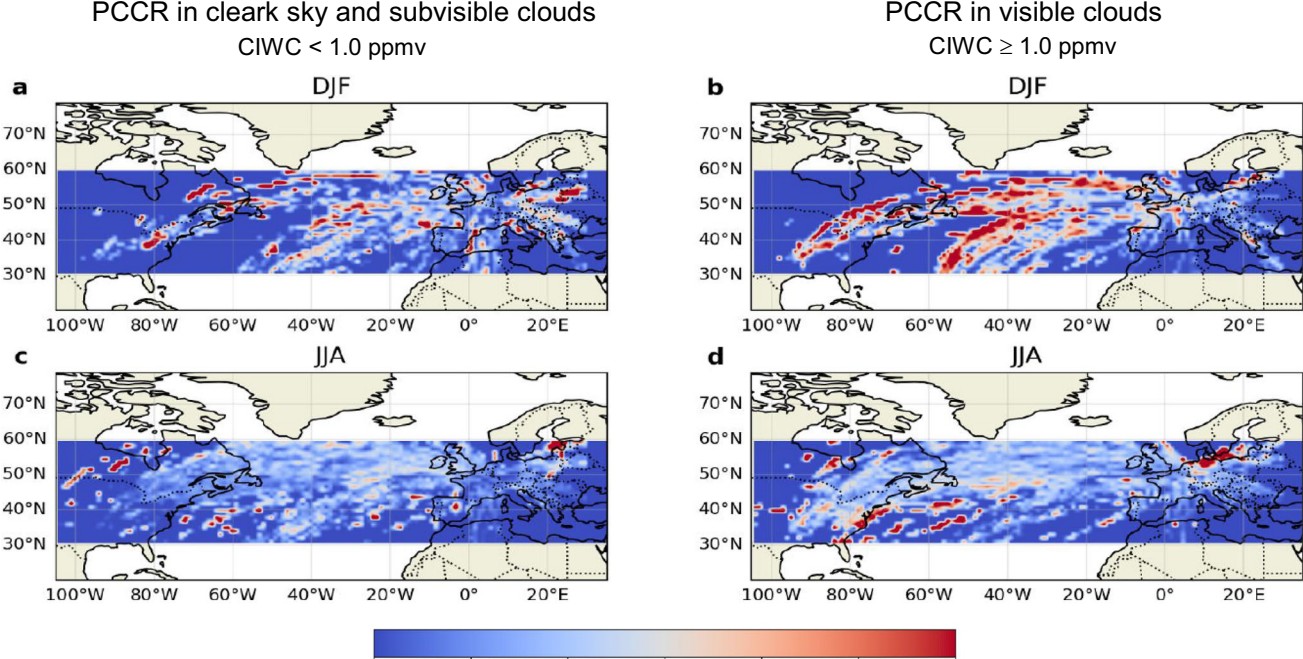

**Fig. 6 | Distribution maps of areas of net warming and ambiguous climate impacts in the Northern midlatitudes.** Seasonal distribution of areas of contrail-cirrus coverage with warming climate impact (**a**, **c**: contrail-cirrus in clear sky or in subvisible cirrus with cloud ice water content CIWC < 1.0 ppmv), and ambiguous climate impact (**b**, **d**: contrail-cirrus inside visible and optically thick cirrus with CIWC ≥ 1.0 ppmv). The maps were determined for potential contrail-cirrus regions (PCCR) with a threshold relative humidity with respect to ice of 90%.

platinum resistance temperature sensor Pt100, is calibrated in the laboratory against an MBW dew-point chilled mirror analyser for $RH_{liq}$[30,59]. A comparison of the ICH with the research-grade FISH fluorescence hygrometer indicates an uncertainty of ± (5-6) % $RH_{liq}$ and ±0.5 K[59].

The along-track cloud ice water content (CIWC) is derived by collocating the 5th version of ECMWF's atmospheric reanalysis meteorological fields, ERA5[60], to IAGOS flight trajectories. The interpolation of ERA5 data at flight positions was conducted at a downgraded spatial and temporal resolution of 1°×1° longitude/latitude resolution and 6 h time resolution. The 137 levels in the vertical from the surface to 0.01 hPa remained unchanged. The CIWC at $T_{air}$ < 235 K is treated as ice water content and serves as the cloud indicator, given that supercooled liquid water droplets freeze instantaneously into ice crystals at temperatures below 235 K. Along-track CIWC, air temperature $T_{ERA5}$ and specific humidity SH are extracted using a weighted-mean method, i.e., selecting the ERA5 grid points that are temporally and spatially closest to the IAGOS observations and calculate the mean values based on their proximity. Additionally, the WMO thermal tropopause height[48] is calculated along flight tracks to distinguish between upper tropospheric and lower stratospheric air masses.

### ERA5 CIWC−based cloud categorisation

To our best knowledge, the ERA5 CIWC has not yet been applied as a cloud indicator to field observations, apart from climatological comparisons with satellite observations[61], or a machine-learning approach towards adjusting ERA5 humidity data to IAGOS observations[62]. In this study, we establish an appropriate CIWC threshold to indicate in-cloud conditions. This is achieved by evaluating ERA5 CIWC against a comprehensive climatology of CIWC data from in-situ observations of total

water and gas phase water (CIWC detection limit 0.001 ppmv)[23], and from cloud particle measurements with the NIXE-CAPS[21] (CIWC detection limit 0.05 ppmv)[63] in 24 field campaigns across the globe[23]. We validate the selected threshold by the in-situ measured $RH_{ice}$ distributions from IAGOS as described in the following.

Panel (a) of Fig. 7 displays the probability distribution of ERA5 $RH_{ice}$ in relation to various ERA5 CIWC thresholds or for the separation of clear-sky and in-cloud conditions at cruise altitude (< 350 hPa and $T_{amb}$ < 235 K) with $RH_{ice}$ and CIWC collocated to research aircraft flight trajectories during the field campaign ML-CIRRUS, one of the 24 field campaigns[20,22], conducted over Central and Western Europe and the Northeast Atlantic. The blue curve, representing ERA5 CIWC = 0.000 ppmv illustrates ideal clear-sky conditions, where ice-supersaturation is rarely observed. As the CIWC threshold for in-cloud categorisation increases to 0.001, 0.005, and finally 0.05 ppmv, the frequency of ice-supersaturation gradually rises due to the inmixing of moist cloud air, and the "shoulder" of the $RH_{ice}$ distribution function around $RH_{ice}$ = 100% develops, which is a clear sign of the presence of ice clouds.

This 'shoulder' is also found in the measurements: in theory, regions in which NIXE-CAPS detects no cloud particles should be classified as cloud-free. However, the noticeable local maximum in the 80−120% $RH_{ice}$ range (black curve in Fig. 7a) for these air masses indicates that the NIXE-CAPS, like every cloud instrument, missed detecting the presence of few small cloud particles in the sampled air which caused a low CIWC < 0.05 ppmv in these nominally cloud-free air masses. When the ERA5 CIWC threshold is raised to 0.05 ppmv (purple curve in Fig. 7a), again the occurrence of ice supersaturation significantly increases compared to lower CIWC thresholds.

When applying the CIWC threshold value of 0.001 ppmv from the in-situ observation of cloud ice water from the total and gas phase

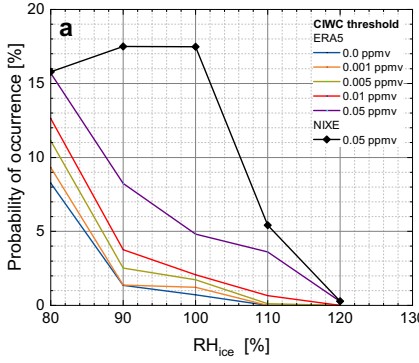
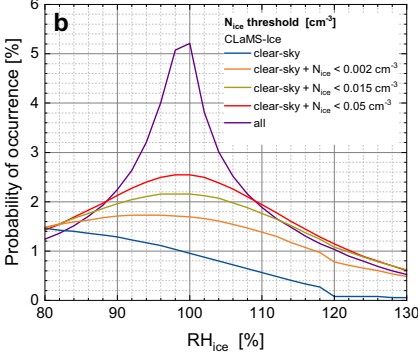

**Fig. 7 | Validation of Cloud Ice Water Content threshold values separating clear-sky from in-cloud environments. a**, occurrence frequency distributions of ERA5[60] relative humidity with respect to ice (RH$_{ice}$) at various ERA5 cloud ice water content (CIWC) thresholds and for NIXE-CAPS observations with the lower detection limit of 0.05 ppmv; **b** RH$_{ice}$ occurrence frequencies at various ice crystal number concentration (N$_{ice}$) thresholds simulated by CLaMS-Ice[64]; all distributions are based on the ML-CIRRUS field campaign conducted over Central and Western Europe and the Northeast Atlantic[21,23].

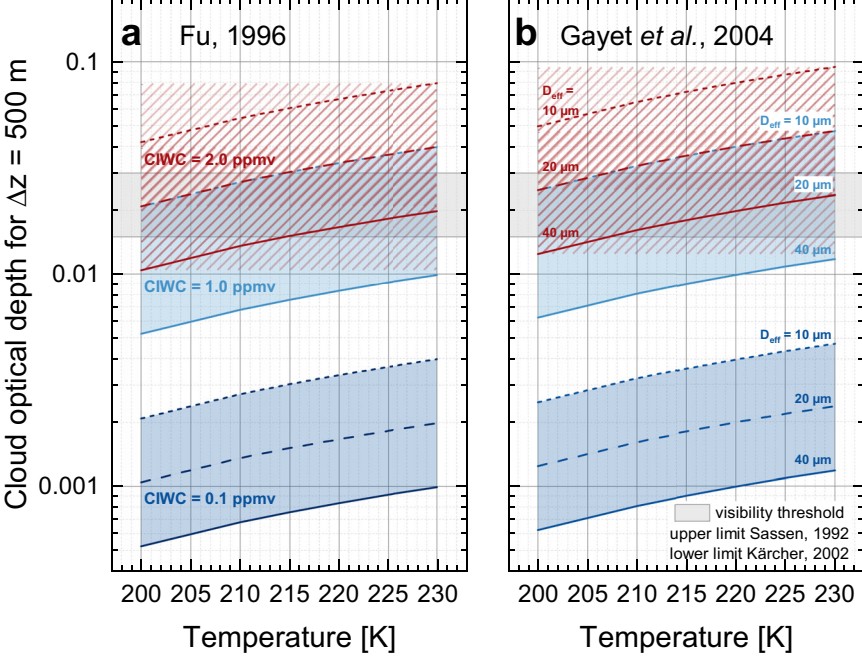

**Fig. 8 | Cloud optical depth thresholds for cloud visibility.** Cloud optical depth thresholds for various ERA5[60] cloud ice water content (CIWC) values and ice crystal effective diameters (D$_{eff}$) as a function of air temperature; curves shown in (**a**) used the radiative scheme by Fu[65], those shown in (**b**) the scheme by Gayet et al.[66].

water measurements of the climatology[23] to the RH$_{ice}$ distributions shown in Fig. 7a (orange curve), the 'shoulder' vanishes and the RH$_{ice}$ PDF is similar to that of clear sky (blue curve), indicating that no clouds were mixed into the air volume. Thus, we adopt an ERA5 threshold CIWC$_{clear}$ = 0.001 ppmv as an indicator of cloud presence in the reanalysis data.

The relation between a local maximum in the RH$_{ice}$ distribution function around 100% and the presence of ice crystals is validated by simulating the occurrence frequencies of RH$_{ice}$ at different thresholds of ice crystal number concentrations, using the Ice module within the Chemical Lagrangian Model of the Stratosphere (CLaMS-Ice)[64]. These simulations are based on ML-Cirrus flight trajectories. CLaMS-Ice employs a two-moment ice microphysics scheme, incorporating both heterogeneous and homogeneous freezing mechanisms, ice crystal growth, evaporation and sedimentation. The model runs 24 h trajectories ending at grid points distributed over the 3D region probed during ML-CIRRUS[64], with boundary conditions provided by reanalysis data. Clear-sky air masses are

defined as those that did not encounter cloud formation along the trajectories. As shown in Fig. 7b, increasing the threshold of the ice crystal number concentration from 0.002 cm$^{-3}$ to 0.05 cm$^{-3}$ enhances the fraction of ice supersaturation, with the typical 'shoulder' emerging at RH$_{ice}$ ≅ 100%. The 'shoulder' is most pronounced when all clouds are included in the RH$_{ice}$ distribution (Fig. 7b, purple curve).

## Cloud optical depth thresholds

To estimate the cloud optical depth from the cloud ice water content, we used the relationships between cloud optical depth, CIWC and ice crystal effective diameter (D$_{eff}$) as provided by Equation (3.9a) of Fu[65] and by Equation (3) of Gayet[66], and assumed a vertical extension of the cloud of 500 m, which is a typical extension for a fully developed contrail[67]. The conversion of CIWC from ppmv into g m$^{-3}$ was conducted for fixed temperature values (in Kelvin) and a related mean pressure inside cirrus clouds following the empirical relationship $p_{mean}$ = 2.89319 × 10$^{-4}$ × T$^{2.66906}$ - 247.724 hPa[68].

Effective diameter values $D_{eff}$ were taken from sources based on observations, from which we selected $D_{eff} = 10\,\mu m$ for a contrail at $T = 217\,K$ and older than 30 min[69], $D_{eff} = 20\,\mu m$ for a young cirrus at 10.6 km altitude and $T = 213\,K$[69], and $D_{eff} = 40\,\mu m$ as the most probable effective diameter for cirrus clouds with CIWC values between 0.1 ppmv and 1.0 ppmv at Northern Midlatitudes[26,70].

The discrimination threshold between subvisible and visible cirrus was taken from two sources. From lidar observations[52] an overall optical depth in the visible spectral range of 0.03 is generally accepted as visibility threshold. From theoretical calculations[53], a limiting extinction coefficient of $2$–$3 \times 10^{-5}\,m^{-1}$ is reported, which turns into an optical depth of 0.01 to 0.015 for a cloud of 500 m vertical extension.

The results shown in Fig. 8 indicate that a CIWC between 1.0 ppmv and 2.0 ppmv, a vertical extension of 500 m, and a temperature range from 205 K to 230 K produce a cloud optical depth between the visibility thresholds from lidar observations[52] and theoretical considerations[53], independent of the selected relationship between optical depth, CIWC and $D_{eff}$. Therefore, we used CIWC values of 1.0 ppmv and 2.0 ppmv to distinguish subvisible from visible clouds ($CIWC_{vis}$).

### Final note on data interpretation

Given the nature of in-situ sampling by aircraft, we can only collect information along the flight path, but have no access to information about cloudiness, etc. above or below flight altitude. Knowing that a contrail developing above or below a pre-existing cloud also has a different radiative impact than a contrail forming in a clear sky column, we cannot investigate these effects in our study. Therefore, the reported fractions of contrail-cirrus within clouds should be considered a lower limit of the contrail-cirrus interacting with pre-existing cloudiness.

### Data availability

IAGOS data are available through the IAGOS Data Portal https://doi.org/10.25326/20. ERA5 data are available through the ECMWF Reanalysis v5 (ERA5) website https://www.ecmwf.int/en/forecasts/dataset/ecmwf-reanalysis-v5. The data set containing the results of the conducted analyses is accessible on Zenodo through https://doi.org/10.5281/zenodo.17327847.

### Code availability

The Python codes used for analysing the data and plotting the analysed data are available from the corresponding author upon request.

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

## Acknowledgements

IAGOS data were created with support from the European Commission, national agencies in Germany (BMBF), France (MESR), and the UK (NERC), and the IAGOS member institutions (http://www.iagos.org/partners). The participating airlines (Deutsche Lufthansa, Air France, China Airlines, Hawaiian Airlines, Air Canada, Iberia, Discover, Cathay Pacific) support IAGOS by carrying the measurement equipment free of charge. The data are available at http://www.iagos.fr thanks to additional support from AERIS. P.S. acknowledges support by the DFG within the Transregional Collaborative Research Centre TRR 301 TPChange, project ID 428312742, project B7, by Carl Zeiss Foundation (Project BINARY, grant P2018-02-003) and by the Internal University Research Funding of Johannes Gutenberg University. Y.L. acknowledges support from the SESAR 3 Joint Undertaking under grant agreement No 101114613 (CICONIA) under European Union's Horizon Europe research and innovation programme. The authors like to thank Dr. Christian Rolf for extracting the ice crystal data from CLaMS-Ice simulations for ML-CIRRUS.

## Author contributions

A.P. and M.K. designed the study and coordinated the analyses and interpretation. N.F.K. performed the IAGOS data analyses. The figures in the manuscript were prepared by A.P. and N.F.K. P.S. contributed the calculations of the cloud optical properties. Y.L. evaluated the ERA5 cloud categorisation and prepared the IAGOS water vapour data set used for this study. S.R. performed the quality assessments of the IAGOS water vapour data. A.P. and M.K. wrote the manuscript with assistance from P.S., Y.L., S.R., S.C. and A.W.

## Funding

## Competing interests

The authors declare no competing interests.
