## [Transparent Peer Review file · Nature Communications]

Most long-lived contrails form within cirrus clouds with uncertain climate impact

Corresponding Author: Professor Andreas Petzold

Version 0:

Reviewer comments:

Reviewer #1

(Remarks to the Author)

The authors present data from seven years of airborne in-situ observations of temperature, relative humidity, and ice-crystal number concentration to assess the occurrence of conditions that, in principle, support the formation of short or long-lived contrails. They find that those conditions are often met within existing cirrus clouds and argue that the effective radiative forcing of those contrails in clouds, though highly uncertain, might be qualitatively assessed based on current knowledge. The authors also propose the use of potential contrail-cirrus region (PCCR) as a more inclusive metric compared to ice super-saturated regions (ISSR) which are commonly consulted to estimated contrail formation but hardly seem to be predict accurately.

Overall, the paper focusses on an important topic that is of high relevance: The best conditions for contrail formation are found within clouds. However, we have no estimate of their radiative forcing as this is currently assessed only for line-shape contrails and contrail cirrus. This reviewer believes that the work makes for a good fit in Nature Communications. Nevertheless, the authors could sharpen their message by addressing the comment below.

Major comments:

The authors build their argument on the observation of a high occurrence rate of PCCR. However, it might be worthwhile to add a paragraph as to whether or not an aircraft that flies through a PCRRs in a cirrus cloud automatically forms a contrail. Are there scenarios in which this might not happen? In the context of the later part of the paper, it would be good to know if PCCR in cirrus and (embedded) contrails are synonymous.

The message of the paper isn't ideally represented in the current title. I suggest to simplify to something like "Most contrails form within cirrus clouds with uncertain climate impact." Here is my take-home message.

1. Conditions for forming long-living contrails are fulfilled most often in regions already covered by cirrus.
2. There are almost no studies of the resulting embedded contrails and nothing is known about their radiative effect.
3. This study provides a qualitative assessment of the radiative forcing of embedded contrails, e.g., for considering the potential impact of re-routing.

In that context, I am missing a clearer statement that our observation-based knowledge on contrails within cirrus clouds is severely limited. This would emphasise the relevance of the authors' findings and allow for defining knowledge gaps to be filled in future research.

While reading the manuscript, I had the feeling that the authors are looking at their dataset from a variety of angles and with slight modification. It might be worthwhile to provide a conceptual description or a flowchart of their line of reasoning. I also have the impression that the authors could optimise their (use of) figures and tables. Some information is provided in both figures and the table, which seems redundant. Some information is also nicely conveyed in the text and doesn't seem to necessitate the addition of a corresponding figure. Finally, the layout of the figures could also be homogenised.

While I appreciate the separation between 3 regions in the northern hemisphere, it seems to me that the overall message of the paper might just as well be conveyed by contrasting just two areas – a combination of what's currently shown for mid-latitudes and the one in the tropics. This would simplify most of the figures and focus the presentation on the conclusions.

I do understand the purpose of Figure 4. Given the broader readership of Nature Communications and the fact that large

parts of the figure are not explained in the caption or the text, I recommend, though, to revise Figure 4 to something more of the nature of a conceptual sketch.

It is really hard to extract information from Figure 6. As the assessment of the climate impact is qualitative, why not just mark if a pixel is dominated by strong or weak climate impact? It would be completely fine to me as reader if this was to be done for the annual average with a discussion of seasonal variation restricted to the text, i.e., without an additional figure.

Minor comments:

I suggest to either add a more quantitative summary of the findings in the Abstract or to clearly state that the findings with respect to the radiative effect are the outcome of a qualitative analysis.

Lines 154 to 158: it is not quite clear how the authors got to these numbers.

Please make sure that your numbering of Tables and Figures is consistent.

Reviewer #2

(Remarks to the Author)

This is an important piece of work that should be published, subject to some revisions/addressing of points.

The m/s addresses the occurrence of persistent contrails within existing cirrus clouds on an observational basis with an impressively large database, which is a largely missing but important contribution to the debate over the role of aviation contrails in climate warming, since evidence for climate warming largely originates from either a very limited number of climate models (2) or diagnostic schemes (1) that have many missing processes. Thus, observational evidence is an important line of additional evidence. Moreover, most calculations that result in net warming are made under simple clear-sky conditions.

I think the abstract could be written a little better, to contain the 'hard-hitting' results (notes below).

There is a tendency throughout the m/s to conflate estimating the 'climate impact' with a premise that there is a necessity for re-routing. The primary contribution of the paper is observational evidence and quantification for the circumstances when persistent contrails occur, i.e. clear sky or within natural cirrus, and then investigating their relative importance. The authors additionally and correctly caution that re-routing proposals consider ISS but do not consider the pre-existence of cirrus clouds. While re-routing as a mitigation approach is of relevance the authors should be clear to the reader that there is really not a demonstrated case for the necessity of this (see recent paper of Bickel et al., 2025, which provides an updated model study of contrail cirrus ERF and temperature response [efficacy]). A clearer separation of the two aspects (size of effect; necessity for mitigation) would make the m/s much clearer and the contribution of the observations more prominent.

Detailed notes:

Abstract

L19-20 This is a premature recommendation that ignores that we cannot predict ISS well on the necessary time/space basis for avoidance, although the "...thorough assessment.." is well stated. It also ignores the fact that the authors' own important findings are not fully understood in terms of 'climate impact' – what is the outcome of persistent contrails formed within pre-existing cirrus.

L15 "...which will change..." do you mean "reduce"?

With the authors' consideration of the above, they can 'spend' more of the limited number of abstract words on their own findings? Candidates for this are points such as "...most contrail-cirrus potentially form in regions already covered by natural cirrus" (L149/150). This is stated quantitatively in the abstract but maybe the statements could be combined.

Instead of the rather vague last sentence, could something more hard-hitting be stated along the lines of "The results of the analysis suggest that contrail cirrus of potentially warming significance rarely forms (or "forms under only limited circumstances") and need to be further analysed and verified in terms of their radiative impact" (just a suggestion).

Main text

L27 nitrogen oxide -> nitrogen oxides

L29 My understanding is that ozone depletion from aircraft NO_x would only occur in the mid stratosphere, where current subsonic aircraft do not fly.

L30 please insert "present day" after 60%, since the proportion depends entirely on recent growth rates of aviation (there is a misconception that this is somehow a fixed proportion).

L32 "mostly for contrail cirrus" this is not necessarily true, considering the poorly quantified aerosol cloud interactions. The authors could correctly nuance this with "of the quantified non-CO₂ effects...".

L41-42 It may be helpful to cross reference lifetimes and radiative response from the overview paper of Karcher (2018, same journal), from his Table 1. There is a general misconception that 'persistent contrails' are radiatively important, which Karcher makes clear that they are not (unless they spread). It is only really persistent contrails that have a lifetime >10 min that spread into contrail-cirrus clouds that matter, radiatively. Indeed, the authors' own results (fig 2 show that most contrails formed, are short-lived and consequently are of trivial radiative importance).

L47-48 may need a little rewording? It could be read as implying that contrail cirrus can sometimes exist without the SAC being fulfilled.

L48-50 may need some nuancing as without qualification, it implies that persistent contrails evolving into contrail cirrus form at ISSR <100%. The observed conditions of pre-formed persistent contrails surviving <100% would only seem plausible under the condition that their formation occurs at >100% in order to be persistent (otherwise they would sublimate) but the contrail cirrus thus formed of larger ice crystals is transitioning to sublimation with a limited lifetime, possibly as a result of them dehydrating the surrounding atmosphere and/or the atmosphere warming.

L52-53 This may need some rewording, as the impression is given that ref 7 is supportive of the definitions made here of PCCRs. Ref 7 does not deal with RHice statistics.

L55-58 Similarly (to L48-50), I find this a little problematic: the existence of contrail cirrus from persistent contrails at >90% does not mean that they were formed at <100% (as above). Their existence at <100% is an observation that needs some additional interpretation to make sense of it.

L60-L69 On a re-read, I think there is an argument missing, which is important for the authors' conclusions. They focus on ISS in L69-79, a prerequisite for persistent contrail formation. While ISS is also prerequisite for cirrus formation, NWP's do not predict cirrus clouds well (in space and time, properties) and many challenges remain. If one of the arguments is that re-routing proposals do not consider persistent contrail formation within clouds, it is worth pointing out that predicting cirrus cloud formation and its properties itself is challenging. This further increases the uncertainty of a re-routing outcome based on a NWP forecast and the risk of re-routing failing to produce a positive climate outcome (let alone verification of it...).

L67-68, I think reference 19 would be highly appropriate here, as would be the work that developed from it:
<https://acp.copernicus.org/articles/24/7911/2024/>

I would be hesitant to cite ref 31, since while the reference is supportive of the present authors' statement, it is only partially so, with ref 31 unconvincingly applauding their own model (their Figure 3) as predicting ISS successfully (a dubious conclusion in my view), which their Figure 3 conceals much detail, which is more honestly exposed in the format of Figure 2 of reference 19.

L71-72. It may be helpful to the reader to be more nuanced here, along the lines of mitigating LLGHG (i.e. CO₂), the primary focus of SAF/tech advancements, and SLCFs such as NO_x products/contrails.

L73-74. The citation of "so called big-hits" is made rather uncritically as if this is a well quantified and therefore real phenomenon. While it is an attractive shorthand which has gained traction in the non-scientific community, the quantification of the statistical distribution of contrail cirrus phenomena in relation to RF is very poor and generally points back to one regional study of Ref 34. I think "individually" could be replaced by "allegedly". Also, I think it would be more accurate to say "have been proposed as a strategy".

L77-79 While not taking away from the cited study, this was actually highlighted by the IPCC WGI AR6 in 2021, Chapter 7, see IPCC WGI AR6 Chapter 7 FAQ 7.2.

L71-79 in general; good points but I am missing the fundamental discussion on the size of the contrail cirrus forcing remains under debate and is only poorly quantified (2 climate models) This is present in L81 but I think the presentation of points could be clearer.

L84 missing "when" between "than they"?

L85 The passing mention in a reference list (ref 40) of Tesche et al does the study rather an injustice, since it is the only observational study (that this reviewer knows of) that attempts to quantify the impact in terms of optical thickness, of an aircraft forming a persistent contrail within a cirrus cloud. I would have thought the authors would maybe consider the results (later in the paper) in comparison to their own, and indeed, acknowledging this single prior observational study bolsters the importance of this present study in filling a much-needed gap.

General. The authors may find a few additional references an interesting/useful part of the story. Gierens 2012 appears to be one of the first authors to investigate aspects numerically (<https://doi.org/10.5194/acp-12-11943-2012>, 2012). Singh et al. (2024) make a few useful points in their review that this is a neglected topic (end of section 2.4) (<https://doi.org/10.5194/acp-24-9219-2024>). Marjani et al. (2022) make some remote sensing measurements (<https://doi.org/10.1029/2021GL096173>).

Section 'PCCRs and ISSRs in major air traffic regions'

The authors base their analysis and reinforcement of the importance of PCCRs over ISSRs on heavily trafficked regions.

This may not be for this study, but I would encourage them to look for what happens in very sparsely trafficked regions where they have observations. Is the ratio/occurrence of persistent contrails <100% the same?

L118 “aviation non-CO2 effects”? Do you mean contrail cirrus, rather than generalizing all aviation’s non-CO2 effects?

L125 Referring to Figure 2. An awkward question, I realise but there is no sense of the uncertainties associated with the percentage splits. Can this be addressed at all (maybe in the SI). Further, on Figure 2 – my understanding is that the outcomes (no contrail, short-lived, persistent) are contingent on the constraints of the observations. To say that “no contrails” are formed in <1% of the air masses in that region is surely ‘for the observations made’, not an absolute truth? Am I missing the point here, or does the text need some small/simple adjustment to account for this?

L140 “mere” ? I’m not sure that this is adding clarity – omit? Or is this supposed to be “more”?

L141” Subvisible clouds are distinguished from optically thick clouds by means CIWC < 1.0 ppmv”. This is a critical/foundational assumption for the analysis of the data and its interpretation, so needs a reference; maybe <https://agupubs.onlinelibrary.wiley.com/doi/full/10.1029/2006JD008214> ?

It would be helpful to point forwards in the text that the authors have explored the sensitivity of this assumption.

L140-L145 The concept of “potential climate impact” is introduced here and results given in Table 2 but not the concept is not explained until the discussion (L191). The whole definition is a little problematic. Figure 4 shows this kind of discrimination in a ‘direction of travel’ basis. The only means of getting to a ‘potential climate impact’ in a quantitative sense that allows a sensible comparison in the different sets of circumstances, is a radiative transfer (RT) calculation, which the authors clearly do not do. Thus, we are left with indicators of ‘directions of travel’, except for the most important phenomenon that the authors identify – that most persistent contrails are formed within pre-existing cirrus. This is a complex topic, hardly researched at all, and with an ambiguous outcome. So, my understanding is that for this category, we do not have a clear ‘direction of travel’. That is OK, it should be more clearly acknowledged and a recommendation made as how to address this with future research.

I think the definition/explanation of “potential climate impact” is better placed here, not in the discussion, if such categorization is to be placed in the results. In Table 2, the entry “no” is confusing. I don’t know what this means. The authors omit to tell the reader that “potential climate impact” is a loose approximation. There will be times that, for example, solar zenith angle will reverse findings via radiative transfer calculations. In (related) L195-L196 for the optically thick cirrus cloud, a “minor effect” is cited. This is vague. If the potential outcome is ambiguous, please say so. It could be cooling or minor warming.

L149-150 States: “That means, most contrail-cirrus potentially form in regions already covered by natural cirrus”. Again, it is unclear what the outcome of this will be. If the pre-existing cirrus is ‘optically thick’, additional contrail cirrus MAY further thicken an already optically-thick (naturally cooling) cloud further. But this is not clear unless more is known, and a full RT calculation is undertaken. All this is incredibly important as this could turn ‘conventional wisdom’ on its head about the net warming effect of contrail cirrus, or at the least, provide observational indirect evidence for a much smaller net warming (since most radiative transfer calculations are made under clear sky conditions). But we simply do not know from these observations. There is some (neglected here) further suggestive, if not definitive, evidence from the modelling study of Verma and Burkhardt (ref 39) who state: “Since corrections in ice nucleation are large in clouds with large ice water content, which are likely connected to large optical depth, *we speculate that disregarding the impact of cirrus on contrail formation may possibly underestimate the cooling of the contrail perturbations within optically thicker clouds*. It should be noted that the cirrus optical depth at which a further increase in optical depth, caused by contrail formation, leads to a cooling is dependent on the cirrus ice crystal habit.” (in ref 39’s conclusions). [* to * is this reviewer’s emphasis]

L193 “change” should this be “increase”?

L200 “relevance is rated as limited”...my understanding is that Verma and Burkhardt (ref 39) found that the additional water mass of ice crystals had a negligible effect (in agreement with earlier work of Gierens (2012, see reference above). If I am correct, would it be useful to be clearer as the sentence addresses ‘climate impact’ through to ‘vortex phase’ with little explanation.

L204 “significant effect” please be more explicit.

L219-225 Figure 4. Please be clearer over what you mean by “stronger” and “weaker” impact. By “stronger”, I am interpreting a greater potential climate impact (as you define it), which I interpret as a positive RF. I am less clear over “weaker” – weaker than what? Is it a cooling impact? Or a less warming impact than clear sky conditions/subvisible conditions? As per above, it seems as if adding optical thickness to an already optically thick cloud (which is ‘cooling’) as the negative forcing of the natural cloud, is made ‘more negative’ which then becomes a negative RF effect (as per the definition of RF, i.e. something interfering with the pre-industrial radiation balance).

L227-232. Same comment over “stronger” and “weaker”. I’m not sure how much value this figure is adding, and if short of space, might be relegated to the SI.

L234. Interesting statement. I don’t disagree from the analysis presented, but this offers quite a different perspective of the ‘big hits’ terminology (of which I remain unconvinced).

L235 I disagree. I think that suggesting this is premature. What is more to the point is that all this needs to be better understood before formulating strategies (which may turn out to be pointless). L236 says this better.

L238 “should help minimizing trade-off effects...” I think this is subtly wrong. I think accounting for all this reduces risks of doing the wrong thing. At the moment, given the authors’ results, the outcome of re-routing has just become far more uncertain.

L234-248. This all makes sense, but it would be nice to have a stronger recommendation as to what research would firm this up (ERF vs occurrence conditions). Can this be accounted for/studied in global models, or can some hybrid observational/radiative transfer study be undertaken?

L242-248. This seems like people-pleasing. We do not know - and this paper adds to the evidence – that re-routing is either a sensible or viable option for mitigating aviation’s impacts on climate. As they say, “get off the fence”! More seriously, simply reiterate the importance of these findings for re-routing proposals.

IN CONCLUSION

I think that the findings are of great importance and significance, but I am uncomfortable with their simplistic treatment of “potential climate impact”. This seems relatively clear, based on prior work with RT models/GCM calculations with RT, that clear sky conditions will net warm. Similarly, this may be the case for sub-visible cirrus with low OD. For persistent contrails forming contrail cirrus clouds within pre-existing cirrus, which seems to be the majority of incidence of persistent contrails from this study, then without much more information, we don’t know what the outcome of this will be. This should be stated much more clearly, and provide a recommendation for further work, given that the authors’ findings suggest that we should have an increased concern over what was already known to be a huge uncertainty – formation of persistent contrails within pre-existing cirrus. The title of the article says it all!

Reviewer #3

(Remarks to the Author)

Summary of the manuscript:

“Contrails inside cirrus clouds predominate with uncertain climate impact” by Petzold et al. examines the potential occurrence of contrails, short-lived or persistent, as a function of humidity and ambient cirrus occurrence. Contrails form when the Schmidt-Appleton criterion is met and nominally can persist in ice-supersaturated regions (ISSR) where RH_{ice} > 100%. Recent research shows that contrails can also persist when RH_{ice} > 90%, giving rise to a larger range of humidities and, hence, a greater number of regions (43% more) of potential contrail coverage. Those regions are designated as potential contrail-cirrus regions (PCCRs). Results are presented for both ISSRs and PCCRs.

The motivation for the study is that a certain class of tropospheric conditions could be eliminated from any aircraft rerouting system that might be established to minimize contrail effects. That particular class of conditions is defined as the presence of thick cirrus clouds at planned flight levels. For this class, the radiative forcing induced by the development of contrail cirrus clouds is assumed to be small, or negligible enough to be ignored. Ergo, there is no need to reroute the flight because of contrail effects.

The analysis uses 7 years of IAGOS aircraft RH_{ice} measurements matched with interpolated ERA 5 to determine the ambient cloud conditions at flight level. A threshold of ERA5 CIWC > 0.001 ppmv is used to differentiate between clear and cloudy conditions. To further classify the cloud conditions, a cloud optical depth (OD) of 0.03, equivalent to CIWC = 1.0 ppmv, was used as the threshold differentiating sub-visible from thick cirrus for a given cloudy datapoint. Thus, any preexisting cirrus with cloud OD > 0.03 is assumed to be thick cirrus. Radiative forcing (RF) as a function of cloud OD is characterized using a plot of RF as a function of cirrus OD provided by Ref 11. The plot reproduced from Ref 11 indicates that RF increases with cloud OD up to a value of ~0.85 then drops precipitously for greater cirrus ODs to negative values for OD > 1.0.

The results of the analysis show that for the 4 regions examined, only 20% (ISSRs) - 30% (PCCRs) of the air masses can support contrail cirrus formation. Of these, roughly 64% of the contrails have a weaker impact on RF (climate) and the remaining 36% have a stronger impact. The overall effects vary regionally and seasonally, with minima over SE Asia and maximum effects during spring and winter. The results lead to the recommendation that any plan to adjust flight levels should include model predictions of cirrus locations, so that planes could fly in areas with thick cirrus and height adjustments would only be needed for clear areas and those with sub visible cirrus. The results also suggest that ~90% of contrails form within thick cirrus over SE Asia, it would probably be of little benefit to undertake any flight rerouting.

Review:

The uncertainty in radiative forcing by contrails is high, leading to low confidence in our understanding of this somewhat small component of the climate change equation. Rerouting aircraft to avoid forming contrail cirrus is probably the most straightforward solution to minimizing contrail effects. However, it involves expenditure of more fuel and would further complicate air traffic control procedures. Thus, minimizing the number of possible aircraft reroutes by eliminating suspect air masses is a worthy effort. Thus, the goal of this study is of interest to the aviation and climate communities.

The premise is logical: contrails forming in a clear sky or in sub visible cirrus should have greater radiative impacts than those forming in thick cirrus. Persistent contrail development in already thick cirrus adds little to either the LW or SW impact of the existing cirrus because of the small change in OD for optically thick cirrus clouds. Yet, there are some shortcomings in how this premise is addressed.

1) The first issue is that the authors looked to no later than 2010 (ref 29) for information about contrail cloud overlap. There are observations of persistent contrail coverage available for 2006 and 2012 (Duda et al., 2013; 2019).

Duda et al. (2013) combined contrail analyses of 1 year of Northern Hemisphere Aqua MODIS data with MODIS cloud retrievals (Minnis et al. (2011)). Using these, Bedka et al. (2013) differentiated between contrails over/within ice clouds (cirrus) and contrails detected in clear skies or over low-level water clouds. Passive satellite retrievals have difficulty detecting cirrus with $OD < 0.05$. The hit rates for clouds having $OD < 0.1$ are around 50% (Trepte et al., 2019). Thus, the clear category would include both clear and subvisible clouds and the ice clouds would be thick according to the definition used here. With that in mind, Bedka et al. (2013) found that contrails were detected with ice clouds predominating in 33% of the cases, compared to 46% with low liquid water clouds, and 17% in clear skies. Seasonal variations are superimposed on those averages. There are likely a few more percent contrails having ice cloud concurrence as ~20% of clouds classified as water were likely to consist of non-opaque cirrus over lower liquid water clouds and a few more water clouds were misclassified as ice than ice clouds misclassified as water (Sun-Mack et al., 2024). Nevertheless, the observed fraction of weaker impact conditions (33% coincidence with ice clouds) is significantly smaller than the 64% computed here. Perhaps, the selection of $CIWC = 0.001$ ppmv as the cloud threshold produces too many clouds. It is not all that different from 0.000 ppmv in frequency. This would be consistent with Reference 57, which indicates that ERA5 generates too much ice cloud in the upper troposphere relative to CALIPSO.

Spangenberg et al. (2013; S13) computed RF for all of the contrails. The resulting mean values support the premise of this paper. Indeed, one could look at those averages and come to the same conclusion as the results of this paper: Flying in an atmosphere with preexisting visible cirrus will cause less RF than flying in other conditions. In fact, the observations actually quantify the differences. S13 found that the net unit contrail RF during the daytime (~13:30 LT) was 3.9 Wm^{-2} for clear skies and 1.3 Wm^{-2} for contrails in the same column as ambient cirrus clouds. That is a factor of 3 decrease, a change that would support the use of the sub visible threshold. Yet, the LW RFs for clear and cloudy scenes are 18.5 and 11.7 Wm^{-2} , respectively, a difference of only 36%. That LW difference is the same at night when no SW RF occurs to balance the LW RF.

2) The premise of this paper seems too limited in scope. One would expect that a contrail developing above or below a thick cirrus would also have a similar impact as a contrail imbedded in the cirrus since the ice cloud OD added to the total column is nearly the same in both cases. Perhaps, that is implied here somehow.

3) The somewhat vague notion of weaker and stronger impacts is not particularly satisfying. Given the observed RF results, what average difference in RF is considered to be enough to separate weak and strong impacts? Is the use of cloud $OD > 0.03$ as a thick cirrus optimal? Is the characterization "thick" warranted or should visible be used always. Normally one would consider a cirrus cloud to be thick when the emissivity approaches unity or the change in emissivity with unit optical depth is relatively small, typically when OD is between 3 and 6. While it is true that adding a contrail to sub visible cirrus will have nearly the same impact as adding a contrail to a clear-sky, there could still be significant contrail RF impacts when the extant cirrus has an $OD < 1.5$ (e.g., Meerkötter et al. 1999). Hong and Liu (2015) show that a large portion of ice clouds observed by CALIPSO have ODs between 0.05 and 1.5, while the mode of the total distribution is $OD = 1.2$. Thus, there will be a range of significant RFs that could be produced by contrails. Determining the optimal cirrus OD threshold would require a sensitivity study and possibly consideration of low-level clouds.

This need for quantification is more apparent when considering that the weaker impacts constitute almost 100% of persistent contrail diagnoses in SE Asia. That suggests that overall RF should be smaller in that region compared to North America or Europe. The RF results from S13 suggest otherwise. They suggest that over the SE Asia flight corridors are much like their USA and European counterparts, and even greater at night.

Other comments

It should be noted that the RF results in Fig. 4 are based on a single set of conditions, noon at $SZA = 50$ deg?. There is no consideration for the ERF at night, which is all warming. The results for other angular conditions, latitudes, and backgrounds will be highly variable.

Line 331 red curve is for 0.01, not 0.05. Purple curve is for 0.05 in Fig. &a.

Despite the issues and reservations described above, not to mention the uncertainties in current model representations of RH_{ice} and clouds, the authors have laid the groundwork for a reasonable method to define cirrus cloud conditions that could theoretically be used to reduce the magnitude of flight rerouting in a future contrail avoidance system.

References

Bedka, S. T., P. Minnis, D. P. Duda, T. L. Chee, and R. Palikonda, 2013: Properties of linear contrails in the Northern

Hemisphere derived from 2006 MODIS observations. *Geophys. Res. Lett.*, 40, 772-777, doi:10.1029/2012GL054363.

Duda, D. P., P. Minnis, K. Khlopenkov, T. L. Chee, and R. Boeke, 2013: Estimation of 2006 Northern Hemisphere contrail coverage using MODIS data. *Geophys. Res. Lett.*, 40, doi:10.1002/grl.50097, 612-617.

Duda, D. P., S. T. Bedka, P. Minnis, D. Spangenberg, K. Khlopenkov, T. Chee, and W. L. Smith, Jr., 2019: Analysis of interannual variability in Northern Hemisphere contrail properties from Terra and Aqua data. *Atmos. Chem. Phys.*, 19, 5313-5330, doi:10.5194/acp-19-5313-2019.

Hong, Y. And G. Liu, 2015: The characteristics of ice cloud properties derived from CloudSat and CALIPSO measurements. *J. Climate*, 28, 3880- 3901. Doi:10.1175/JCLI-D-14-00666.1

Meerkötter, R., U. Schumann, D. R. Doelling, P. Minnis, T. Nakajima, and Y. Tsushima, 1999: Radiative forcing by contrails, *Annales Geophysicae*, 17, 1070-1084.

Minnis, P., S. Sun-Mack, D. F. Young, P. W. Heck, D. P. Garber, et al., 2011: CERES Edition-2 cloud property retrievals using TRMM VIRS and Terra and Aqua MODIS data, Part I: Algorithms. *IEEE Trans. Geosci. Remote Sens.*, 49, 11, 4374-4400, doi:10.1109/TGRS.2011.2144601.

Spangenberg, D. A., P. Minnis, S. T. Bedka, R. Palikonda, D. P. Duda, and F. G. Rose, 2013: Contrail radiative forcing over the Northern Hemisphere from 2006 Aqua MODIS data. *Geophys. Res. Lett.*, 40, doi:10.1002/grl.50168, 595-600.

Trepte, Q. Z., P. Minnis, S. Sun-Mack, C. R. Yost, Y. Chen, Z. Jin, F.-L. Chang, W. L. Smith, Jr., K. M. Bedka, and T. L. Chee, 2019: Global cloud detection for CERES Edition 4 using Terra and Aqua MODIS data. *IEEE Trans. Geosci. Remote Sens.*, 57, 9410-9449, doi:10.1109/TGRS.2019.2926620.

Sun-Mack, S., P. Minnis, Y. Chen, G. Hong, and W. L. Smith, Jr., 2024: Identification of ice-over-water multilayer clouds using an artificial neural network with multispectral satellite data. *Atmos. Meas. Tech.*, 17, 3323–3346, doi:10.5194/amt-17-3323-2024.

Version 1:

Reviewer comments:

Reviewer #1

(Remarks to the Author)

The authors have carefully revised their manuscript according to the comments provided by three reviewers. The paper is now very straightforward to read and conveys a clear message. The use of Figure 1 for illustrating the conceptual expectation is good.

Going through the manuscript, I found it difficult to reconcile the information provided in Figures 3 and 5, which are clearly showing different combinations of the same numbers. While all became clear after inspecting Table 1, I would try to avoid similar confusion with readers by omitting Figure 5 from the paper.

Overall, this is great and important work that sheds some light on an understudied phenomenon. I am happy to see this published.

Reviewer #2

(Remarks to the Author)

The authors have responded comprehensively and thoroughly to all three reviews and I recommend publication as soon as possible of the ground-breaking paper.

I would only suggest one tiny modification, where the authors have (I think) possibly misunderstood my point. My comment was "L30 please insert "present day" after 60%, since the proportion depends entirely on recent growth rates of aviation (there is a misconception that this is somehow a fixed proportion)."

The authors replied: We changed the sentence to "With an ERF of 60 mW m⁻² contrail-cirrus is estimated to be the largest non-CO₂ ERF contributing 60% (present-day knowledge) to the aviation net total ERF of 100 mW m⁻²".

I was not referring to "present-day knowledge" (and proportions, thereof) of aviation ERF assessments but rather that the proportion of non-CO₂ forcing to the total is a function of the history of CO₂ emissions and the complexities of C cycle CO₂ accumulation (and subsequent ERF) and the simpler linear response of SLCFs of ERF to emissions (in general). This can be seen in Figure 2c of Klower et al. (2021) <https://doi.org/10.1088/1748-9326/ac286e> (RH bars), where different growth rate scenarios result in different proportions of response (it's actually dT but that matters little to the point being made).

The authors do not need to add a reference or anything like that, I only point out to clarify and they can very simply fix this by removing the word "knowledge".

Otherwise, great job!

Reviewer #3

(Remarks to the Author)

The authors have satisfactorily answered my concerns and altered the paper accordingly. Thus, would recommend publication in its current form.

POINT-BY-POINT RESPONSE TO REVIEWER COMMENTS

The authors gratefully acknowledge the thorough and constructive reviews by the three anonymous referees. All points raised were carefully considered and taken up in the revision of the manuscript.

As recommended by the referees, we strengthened the focus on the presentation and discussion of our observational results, rewrote the entire abstract, and concentrate the discussion of results on the topic of contrail-cirrus appearance in clear sky and subvisible as well as visible cirrus.

We describe with more detail the approach of our study which is now presented in Figure 1. The ‘climate impact’ concept is replaced by the concept that contrail-cirrus in clear sky and subvisible cirrus causes considerable warming while the warming effect of contrail-cirrus in visible clouds is not only minor but also ambiguous since, depending on the optical thickness of the pre-existing cirrus, the effect can shift from warming to cooling. Thanks to the referees’ comments, we are now able to quantify the enhancement of warming of contrails in clear sky and subvisible cirrus compared to visible clouds (Spangenberg et al., 2013; Bedka et al., 2013), and support the potential cooling of contrail-cirrus embedded in thick cirrus clouds by a reference to a model study (Verma and Burkhardt, 2022). In consequence categories ‘stronger climate impact’ and ‘weaker climate impact’ are replaced by the categories ‘warming’ and ‘ambiguous’.

To make the results of our study more visible, we followed the recommendation of Reviewer 1 and merged the former regions 1- 3 into one region on Northern Midlatitudes which is then directly compared to the Southeast Asian Subtropics. The details for former regions 1 – 3 are added to the figures to quantify the observed variability.

Since this is a very active research field, we updated the manuscript with several publications which appeared recently (Zhang et al., 2025; Bickel et al., 2025; Prather et al., 2025). Also, the extensive review by Singh et al. (2024) on the role of contrails and contrail cirrus in climate change was added.

For each referee, we will respond to the questions and recommendations point by point. Our answers are shown in italic green. Please note that we did not include the references in the text excerpts we added to the response to reviewers. We also provide an annotated manuscript showing the modifications in track-changes-mode. Line numbers refer to the clean revised manuscript.

Reviewer #1 (Remarks to the Author):

The authors present data from seven years of airborne in-situ observations of temperature, relative humidity, and ice-crystal number concentration to assess the occurrence of conditions that, in principle, support the formation of short or long-lived contrails. They find that those conditions are often met within existing cirrus clouds and argue that the effective radiative forcing of those contrails in clouds, though highly uncertain, might be qualitatively assessed based on current knowledge. The authors also propose the use of potential contrail-cirrus region (PCCR) as a more inclusive metric compared to ice super-saturated regions (ISSR) which are commonly consulted to estimated contrail formation but hardly seem to be predict accurately.

Overall, the paper focusses on an important topic that is of high relevance: The best conditions for contrail formation are found within clouds. However, we have no estimate of their radiative forcing as this is currently assessed only for line-shape contrails and contrail cirrus. This reviewer believes that the work makes for a good fit in Nature Communications. Nevertheless, the authors could sharpen their message by addressing the comment below.

Major comments:

The authors build their argument on the observation of a high occurrence rate of PCCR. However, it might be worthwhile to add a paragraph as to whether or not an aircraft that flies through a PCRRs in a cirrus cloud automatically forms a contrail. Are there scenarios in which this might not happen?

Reply: Very good point, we added the following paragraph to the manuscript on line 80 to 84:

“Aircraft flying through an ice-supersaturated or even slightly subsaturated air mass will thus form a persistent contrail and finally a contrail-cirrus as soon as ambient air temperature is below 230 - 225 K, which is almost always the case for Northern midlatitude cruise altitude. Only for warmer temperatures above 230 K, as met, e.g., in the subtropics or at lower altitudes in the midlatitudes, these regions may be crossed by commercial aircraft without generating a contrail. “

In the context of the later part of the paper, it would be good to know if PCCR in cirrus and (embedded) contrails are synonymous.

Reply: Indeed, PCCR in cirrus and embedded contrails are synonymous. This is clarified on line 94.

The message of the paper isn't ideally represented in the current title. I suggest to simplify to something like “Most contrails form within cirrus clouds with uncertain climate impact.”

Reply: Thank you very much for this excellent suggestion, we modified the title accordingly but added ‘long-living’ to be precise.

Here is my take-home message.

1. Conditions for forming long-living contrails are fulfilled most often in regions already covered by cirrus.
2. There are almost no studies of the resulting embedded contrails, and nothing is known about their radiative effect.
3. This study provides a qualitative assessment of the radiative forcing of embedded contrails, e.g., for considering the potential impact of re-routing.

Reply: We are grateful to the reviewer for this stringent formulation of our take-home message. We adopted the wording in our abstract and conclusion sections which reads now:

“The results of our study suggest that conditions necessary for the formation of contrail-cirrus are most often fulfilled in regions that are already covered by visible cirrus, which reduces their warming impact or even turns it into a cooling effect. Those of a stronger warming significance, however, form under only limited circumstances in clear sky and subvisible cirrus. Unfortunately, our knowledge on the radiative effects of contrail-cirrus embedded in subvisible, visible and optically thick clouds, and feedback mechanisms between contrail and cirrus ice particles is highly limited. There will be a range of radiative forcings than could be produced by embedded contrail-cirrus, and the quantification of the climate impacts would require a sensitivity study and possibly consideration of low-level clouds.

This topic needs to be further analysed and verified in terms of their radiative impact for consideration of the potential impact of contrail avoidance strategies including flight rerouting. Finally, ice-supersaturation only is by far not a reliable, stand-alone criterion for assessing contrail-cirrus climate effects. Instead, we need to include pre-existing cirrus clouds and their interactions with contrail-cirrus into all our considerations.”

In that context, I am missing a clearer statement that our observation-based knowledge on contrails within cirrus clouds is severely limited. This would emphasise the relevance of the authors’ findings and allow for defining knowledge gaps to be filled in future research.

Reply: Indeed, a clearer statement is required, we added a sentence to the introduction part, starting at line 60:

“Observation-based knowledge on long-lived contrails within cirrus clouds and their radiative effects is severely limited. To narrow that knowledge gap, we study the occurrence of conditions favouring the formation and existence of long-lived contrails inside pre-existing cirrus clouds from observations of relative humidity with respect to ice taken on board of passenger aircraft equipped with suitable instrumentation. “.

While reading the manuscript, I had the feeling that the authors are looking at their dataset from a variety of angles and with slight modification. It might be worthwhile to provide a conceptual description or a flowchart of their line of reasoning.

Reply: We fully agree and start now the results section with a description of our conceptual approach. We refer the reviewer to the manuscript from line 134 to line 159.

I also have the impression that the authors could optimise their (use of) figures and tables. Some information is provided in both figures and the table, which seems redundant. Some information is also nicely conveyed in the text and doesn’t seem to necessitate the addition of a corresponding figure.

Reply: Figures and table have been carefully revisited and adapted where necessary. We have paid particular attention to the potential redundancy between figures and table. Although Table 1 still contains some redundancy, we decided to keep it because we believe it is of good value to give numbers of cloud fractions etc. in addition to graphical presentation.

Finally, the layout of the figures could also be homogenised.

Reply: Figures have been homogenised, as recommended.

While I appreciate the separation between 3 regions in the northern hemisphere, it seems to me that the overall message of the paper might just as well be conveyed by contrasting just two areas – a combination of what’s currently shown for mid-latitudes and the one in the tropics. This would simplify most of the figures and focus the presentation on the conclusions.

Reply: Very good suggestion, figures have been modified accordingly. Since we still wanted to keep the North Atlantic region marked, we added the analyses for the separate regions of the Northern Midlatitudes as smaller panels to the Figures 3 and 4. Figure 5 on the climate impact is now shown only for the entire Northern midlatitudes and the Southeast Asian subtropics. Detailed numbers for all regions are shifted to Tables S1 and S2 in the supplement.

I do understand the purpose of Figure 4. Given the broader readership of Nature Communications and the fact that large parts of the figure are not explained in the caption or the text, I recommend, though, to revise Figure 4 to something more of the nature of a conceptual sketch.

Reply: Figure 4 has been revised to a conceptual figure, as suggested. A detailed description has been added to the first paragraph of the discussion section. Former Figure 4 is now put at the front of the results section as Figure 1 to guide the reader through the concept. We refer the reviewer to lines 142 to 159.

It is really hard to extract information from Figure 6. As the assessment of the climate impact is qualitative, why not just mark if a pixel is dominated by strong or weak climate impact? It would be completely fine to me as reader if this was to be done for the annual average with a discussion of seasonal variation restricted to the text, i.e., without an additional figure.

Reply: Figure 6 have been simplified and shows now distribution patterns for summer and winter seasons only. The resolution was reduced so that the appearance is now “blurrier” which makes the lateral distribution easier to recognise. Nevertheless, we want to keep this information in the manuscript since it shows the regional and seasonal variabilities for the net warming and ambiguous impact cases.

Minor comments:

I suggest to either add a more quantitative summary of the findings in the Abstract or to clearly state that the findings with respect to the radiative effect are the outcome of a qualitative analysis.

Reply: We added the following sentence to the abstract: “A conceptual analysis shows that subvisible cirrus and clear-sky cover ~10% of the cruise altitude over Northern midlatitudes (< 2% in the subtropics) and contrails within these regions are expected to cause additional warming. However, most contrails in the thicker, visible cirrus, only slightly enhance the cirrus warming effect or possibly reverse it to cooling. Our results suggest that potential flight rerouting concepts for contrail avoidance need to consider cirrus cloud coverage in addition to ice-supersaturation.

Lines 154 to 158: it is not quite clear how the authors got to these numbers.

Reply: The entire section on the overlap of regions covered by cirrus clouds and prone to contrail formation has been revised. The new section on the overlap between contrail-cirrus and visible cirrus is now from line 230 to line 245. For all numbers we clearly indicate where the numbers are taken from or how they have been calculated.

Please make sure that your numbering of Tables and Figures is consistent.

Reply: The numbering of Tables and Figures has been carefully checked and modified where necessary.

Reviewer #2 (Remarks to the Author):

This is an important piece of work that should be published, subject to some revisions/addressing of points.

The m/s addresses the occurrence of persistent contrails within existing cirrus clouds on an observational basis with an impressively large database, which is a largely missing but important contribution to the debate over the role of aviation contrails in climate warming, since evidence for climate warming largely originates from either a very limited number of climate models (2) or diagnostic schemes (1) that have many missing processes. Thus, observational evidence is an important line of additional evidence. Moreover, most calculations that result in net warming are made under simple clear-sky conditions.

I think the abstract could be written a little better, to contain the ‘hard-hitting’ results (notes below).

There is a tendency throughout the m/s to conflate estimating the ‘climate impact’ with a premise that there is a necessity for re-routing. The primary contribution of the paper is observational evidence and quantification for the circumstances when persistent contrails occur, i.e. clear sky or within natural cirrus, and then investigating their relative importance. The authors additionally and correctly caution that re-routing proposals consider ISS but do not consider the pre-existence of cirrus clouds. While re-routing as a mitigation approach is of relevance the authors should be clear to the reader that there is really not a demonstrated case for the necessity of this (see recent paper of Bickel et al., 2025, which provides an updated model study of contrail cirrus ERF and temperature response [efficacy]). A clearer separation of the two aspects (size of effect; necessity for mitigation) would make the m/s much clearer and the contribution of the observations more prominent.

Reply: Fully agreed, see comment above on the overall structure of the study. Furthermore, the abstract has been completely rewritten. We refer the reviewer to the revised abstract.

Detailed notes:

Abstract

L19-20 This is a premature recommendation that ignores that we cannot predict ISS well on the necessary time/space basis for avoidance, although the “...thorough assessment.” is well stated. It also ignores the fact that the authors’ own important findings are not fully understood in terms of ‘climate impact’ – what is the outcome of persistent contrails formed within pre-existing cirrus.

Reply: This recommendation is removed from the abstract, and the entire abstract was rewritten with a stronger focus on the quantitative results we report.

L15 “...which will change...” do you mean “reduce”?

Reply: The original wording was indeed not precise, and we followed the reviewer’s suggestion and use the term “reduce” instead.

With the authors’ consideration of the above, they can ‘spend’ more of the limited number of abstract words on their own findings? Candidates for this are points such as “...most contrail-cirrus potentially form in regions already covered by natural cirrus” (L149/150). This is stated quantitatively in the abstract but maybe the statements could be combined.

Instead of the rather vague last sentence, could something more hard-hitting be stated along the lines of “The results of the analysis suggest that contrail cirrus of potentially warming significance rarely

forms (or “forms under only limited circumstances”) and need to be further analysed and verified in terms of their radiative impact” (just a suggestion).

Reply: As said above, the entire abstract was rewritten with a stronger focus on the quantitative results we report. However, we decided to keep a reference to the rerouting discussion in the abstract which reads ‘Our results suggest that potential flight rerouting concepts for contrail avoidance need to consider cirrus cloud coverage in addition to ice-supersaturation.’ The idea behind keeping this sentence is to make readers from the “rerouting business” aware of our results which otherwise might get lost.

Main text

L27 nitrogen oxide -> nitrogen oxides

Reply: Corrected

L29 My understanding is that ozone depletion from aircraft NO_x would only occur in the mid stratosphere, where current subsonic aircraft do not fly.

Reply: This understanding is correct. However, from the REACT4C project there are also cases reported for reduced ozone in the lowermost stratosphere at high latitudes. Since the climate change function of aviation-induced ozone are always positive, we rephrased the statement as: “aviation-NO_x induced ozone formation”.

L30 please insert “present day” after 60%, since the proportion depends entirely on recent growth rates of aviation (there is a misconception that this is somehow a fixed proportion).

Reply: We changed the sentence to “With an ERF of 60 mW m⁻² contrail-cirrus is estimated to be the largest non-CO₂ ERF contributing 60% (present-day knowledge) to the aviation net total ERF of 100 mW m⁻²”.

L32 “mostly for contrail cirrus” this is not necessarily true, considering the poorly quantified aerosol cloud interactions. The authors could correctly nuance this with “of the quantified non-CO₂ effects...”.

Reply: Thank you, we changed this accordingly.

L41-42 It may be helpful to cross reference lifetimes and radiative response from the overview paper of Karcher (2018, same journal), from his Table 1. There is a general misconception that ‘persistent contrails’ are radiatively important, which Karcher makes clear that they are not (unless they spread). It is only really persistent contrails that have a lifetime >10 min that spread into contrail-cirrus clouds that matter, radiatively. Indeed, the authors’ own results (fig 2 show that most contrails formed, are short-lived and consequently are of trivial radiative importance).

Reply: Very good suggestion, indeed. We rephrased the related paragraph as follows (lines 71 – 75):

‘After reaching the ambient temperature, the contrail ice particles grow or shrink in size depending on the ambient humidity. If the environment remains ice-supersaturated ($RH_{ice} > 100\%$), the contrails can persist with lifetimes from 4 to > 10 hours. If the ambient air is ice-subaturated ($RH_{ice} < 100\%$), the ice particles sublime and the contrail dissolves. For a contrail to survive more than approx. 10 minutes and to spread out to radiatively impactful contrail cirrus, ambient RH_{ice} must be close to or above ice-saturation.’

L47-48 may need a little rewording? It could be read as implying that contrail cirrus can sometimes exist without the SAC being fulfilled.

Reply: Agreed, we simply removed the part of the sentence referring to SAC. It reads now (lines 86 – 87): “Today, the occurrence of contrail-cirrus is discussed in its relationship to cold ($T < 230$ K) ice-supersaturated regions (ISSR: $RH_{ice} \geq 100\%$).”

L48-50 may need some nuancing as without qualification, it implies that persistent contrails evolving into contrail cirrus form at ISSR $<100\%$. The observed conditions of pre-formed persistent contrails surviving $<100\%$ would only seem plausible under the condition that their formation occurs at $>100\%$ in order to be persistent (otherwise they would sublime) but the contrail cirrus thus formed of larger ice crystals is transitioning to sublimation with a limited lifetime, possibly as a result of them dehydrating the surrounding atmosphere and/or the atmosphere warming.

Reply: Indeed, this section requires clarification to avoid misunderstanding. We added the following sentences (lines 87 – 92):

“To include the observationally confirmed existence of contrail-cirrus also at slight ice-subsaturation in the consideration of aviation climate impacts, we introduce the term “potential contrail-cirrus region” (PCCR) which we define as an air mass with $RH_{ice} \geq 90\%$ and SAC fulfilled. In such an air mass with SAC fulfilled, a contrail would form and go through the stage of ice-supersaturation during plume expansion where the ice crystals grow. The resulting contrail formed of larger ice crystals would then transition to sublimation with a limited but long-enough lifetime for becoming radiatively impactful.”

L52-53 This may need some rewording, as the impression is given that ref 7 is supportive of the definitions made here of PCCRs. Ref 7 does not deal with RHice statistics.

Reply: Ref 7 was used as a source of information for the lifetime. Since this reference was made earlier, we removed the reference here.

L55-58 Similarly (to L48-50), I find this a little problematic: the existence of contrail cirrus from persistent contrails at $>90\%$ does not mean that they were formed at $<100\%$ (as above). Their existence at $<100\%$ is an observation that needs some additional interpretation to make sense of it.

Reply: The fact that contrail formation and first growth will always go through a stage of ice supersaturation was explained on lines 87 – 92 and should be clear now.

L60-L69 On a re-read, I think there is an argument missing, which is important for the authors' conclusions. They focus on ISS in L69-79, a prerequisite for persistent contrail formation. While ISS is also prerequisite for cirrus formation, NWP's do not predict cirrus clouds well (in space and time, properties) and many challenges remain. If one of the arguments is that re-routing proposals do not consider persistent contrail formation within clouds, it is worth pointing out that predicting cirrus cloud formation and its properties itself is challenging. This further increases the uncertainty of a re-routing outcome based on a NWP forecast and the risk of re-routing failing to produce a positive climate outcome (let alone verification of it...).

Reply: Thank you very much for that excellent suggestion, we added the sentence on lines 112 – 114): “It also needs to be mentioned that forecasting cirrus cloud formation and its properties is challenging for the same reasons as those for ISSRs and potentially PCCRs, since ISSRs are an indispensable prerequisite for cirrus cloud formation.”

L67-68, I think reference 19 would be highly appropriate here, as would be the work that developed from it: <https://acp.copernicus.org/articles/24/7911/2024/>
I would be hesitant to cite ref 31, since while the reference is supportive of the present authors'

statement, it is only partially so, with ref 31 unconvincingly applauding their own model (their Figure 3) as predicting ISS successfully (a dubious conclusion in my view), which their Figure 3 conceals much detail, which is more honestly exposed in the format of Figure 2 of reference 19.

Reply: We fully agree, removed Ref 31, inserted Ref 19 and its update (now 31 and 36) instead.

L71-72. It may be helpful to the reader to be more nuanced here, along the lines of mitigating LLGHG (i.e. CO₂), the primary focus of SAF/tech advancements, and SLCFs such as NO_x products/contrails.

Reply: We extended the argument which reads now (lines 116 – 118):

“There are several measures discussed for reducing aviation’s climate impact, including technological advancements for reducing the emissions of short-lived climate forcers such as NO_x and aerosols, and the availability of sustainable aviation fuels for reducing long-lived greenhouse gases “

L73-74. The citation of “so called big-hits” is made rather uncritically as if this is a well quantified and therefore real phenomenon. While it is an attractive shorthand which has gained traction in the non-scientific community, the quantification of the statistical distribution of contrail cirrus phenomena in relation to RF is very poor and generally points back to one regional study of Ref 34. I think “individually” could be replaced by “allegedly”. Also, I think it would be more accurate to say “have been proposed as a strategy”.

Reply: We fully agree, removed the term “big hit” which is indeed a bit sensational, and modified the sentence as (line 118 – 119): “On the short term, operational concepts for avoiding contrails that allegedly have the largest warming effects, have been proposed as a strategy for reducing aviation’s climate impact “.

L77-79 While not taking away from the cited study, this was actually highlighted by the IPCC WGI AR6 in 2021, Chapter 7, see IPCC WGI AR6 Chapter 7 FAQ 7.2.

Reply: Thank you for the hint, the sentence was rephrased to “Additionally, the latest IPCC report highlighted the role of clouds in a warming climate (IPCC, 2021) which makes the reduction of anthropogenically generated high ice clouds even more urgent.”

L71-79 in general; good points but I am missing the fundamental discussion on the size of the contrail cirrus forcing remains under debate and is only poorly quantified (2 climate models) This is present in L81 but I think the presentation of points could be clearer.

Reply: We added the following sentence and reference to the last sentence of the first paragraph (lines 32 - 36): “Other than the well-understood CO₂ effect and its small uncertainty, the quantified non-CO₂ effects of aviation are associated with large uncertainties, mostly for contrail-cirrus with a substantial associated uncertainty of approx. 70 %. In particular, the feedbacks of contrail-cirrus radiative effects and surface temperature are largely unknown, but a most-recent model study including these effects reports that the climate sensitivity is lower for contrail-cirrus than for aviation-related CO₂ emissions.”.

L84 missing “when” between “than they”?

Reply: Indeed, the sentence is unclear and was rearranged accordingly.

L85 The passing mention in a reference list (ref 40) of Tesche et al does the study rather an injustice, since it is the only observational study (that this reviewer knows of) that attempts to quantify the impact in terms of optical thickness, of an aircraft forming a persistent contrail within a cirrus cloud. I would have thought the authors would maybe consider the results (later in the paper) in comparison

to their own, and indeed, acknowledging this single prior observational study bolsters the importance of this present study in filling a much-needed gap.

Reply: Fully agreed. We discuss now the references Tesche et al. and Marjani et al. (refs 16 and 17) separately and back it up with the recent paper by Zhang et al. (2025) who also report a strong impact on contrail cirrus optical depth on the effective radiative forcing. The section added on line 55 reads “Two studies using CALIPSO observations and flight track recordings reported an increase in cloud optical depth by 22% after an aircraft had crossed the cloud, and a statistically significant increase by as large as a factor of 2 in the concentration of ice crystals in clouds affected by aviation. A most-recent modelling study on the impact of existing uncertainty in contrail cirrus optical depth on the resulting contrail-cirrus ERF reports an eightfold uncertainty.”

General.

The authors may find a few additional references an interesting/useful part of the story. Gierens 2012 appears to be one of the first authors to investigate aspects numerically (<https://doi.org/10.5194/acp-12-11943-2012>, 2012). Singh et al. (2024) make a few useful points in their review that this is a neglected topic (end of section 2.4) (<https://doi.org/10.5194/acp-24-9219-2024>). Marjani et al. (2022) make some remote sensing measurements (<https://doi.org/10.1029/2021GL096173>).

Reply: Thank you very much for these valuable hints. The references have been integrated at the suitable points of the story.

Section ‘PCCRs and ISSRs in major air traffic regions’

The authors base their analysis and reinforcement of the importance of PCCRs over ISSRs on heavily trafficked regions. This may not be for this study, but I would encourage them to look for what happens in very sparsely trafficked regions where they have observations. Is the ratio/occurrence of persistent contrails <100% the same?

Reply: The importance of PCCRs over ISSRs was developed from analyses of dedicated research flights in the field campaigns CONCERT 2008 (Kübbeler et al., 2008) and ML-CIRRUS (Li et al., 2023). Our analysis used the observed evidence for defining PCCRs. Unfortunately, we cannot repeat this type of analysis for low-traffic regions since we don’t have the data.

L118 “aviation non-CO₂ effects”? Do you mean contrail cirrus, rather than generalizing all aviation’s non-CO₂ effects?

Reply: We clarified this point and rephrased the sentence to “have immediate consequences on all aviation non-CO₂ effects in these different regions of the world, and particularly for contrail-cirrus”.

L125 Referring to Figure 2. An awkward question, I realise but there is no sense of the uncertainties associated with the percentage splits. Can this be addressed at all (maybe in the SI).

Reply: Indeed, we cannot add uncertainties to the percentages since these values arise from a simple split of the entire data set into these three fractions. As suggested by Reviewer 1, we merged the three regions of the Northern midlatitudes to one and repeated the analysis. Now, the reader finds the fractional split in Figure 3a and check for the variability arising from the different regions in Figures 3c to 3e. This analysis gives at least some information on the uncertainty in this split. Same was done for

Figure 4 on the distributions of RH_{ice} . Respective numerical information is available in the Supplement in Tables S1 and S2.

Further, on Figure 2 – my understanding is that the outcomes (no contrail, short-lived, persistent) are contingent on the constraints of the observations. To say that “no contrails” are formed in <1% of the air masses in that region is surely ‘for the observations made’, not an absolute truth? Am I missing the point here, or does the text need some small/simple adjustment to account for this?

Reply: Thank you for that hint, we added the suggested phrase “for the observations made” to the sentence on line 198. We also rephrased the next sentence to “Conditions favouring the formation of short-lived contrails (defined as SAC fulfilled, $RH_{ice} < 90\%$) dominate in all regions with fractions of 70% and more of the probed air masses “.

L140 “mere” ? I’m not sure that this is adding clarity – omit? Or is this supposed to be “more”?

Reply: Agreed, we removed the word “mere”.

L141” Subvisible clouds are distinguished from optically thick clouds by means $CIWC < 1.0$ ppmv”. This is a critical/foundational assumption for the analysis of the data and its interpretation, so needs a reference; maybe <https://agupubs.onlinelibrary.wiley.com/doi/full/10.1029/2006JD008214> ? It would be helpful to point forwards in the text that the authors have explored the sensitivity of this assumption.

Reply: The details of our approach for cloud optical depth are described in the Methods section where we also explored the sensitivity of this assumption. To point to the Methods, we modified the sequence starting on line 224 accordingly to: “We applied the CIWC threshold values $CIWC_{vis}$ for cloud visibility of 1.0 ppmv and 2.0 ppmv to quantify the dependence of our results on the doubling of the CIWC visibility threshold value. A statistical analysis of the conditions and the connected climate impact of these three formation categories (clear sky, subvisible, visible) is given in Table 1. Details on the definition of the regions, CIWC categorization, and the results of the full statistical analysis are given in ‘Methods’ and in Table S1 of the supplementary information.

In ‘Methods’, we further explain: “The discrimination threshold between subvisible and visible cirrus was taken from two sources. From lidar observations (Sassen et al., 1992) an overall optical depth in the visible spectral range of 0.03 is generally accepted as visibility threshold. From theoretical calculations (Kärcher, 2001), a limiting extinction coefficient of $2 - 3 \times 10^{-5} m^{-1}$ is reported, which turns into an optical depth of 0.01 to 0.015 for a cloud of 500 m vertical extension.

L140-L145 The concept of “potential climate impact” is introduced here and results given in Table 2 but not the concept is not explained until the discussion (L191). The whole definition is a little problematic. Figure 4 shows this kind of discrimination in a ‘direction of travel’ basis. The only means of getting to a ‘potential climate impact’ in a quantitative sense that allows a sensible comparison in the different sets of circumstances, is a radiative transfer (RT) calculation, which the authors clearly do not do. Thus, we are left with indicators of ‘directions of travel’, except for the most important phenomenon that the authors identify – that most persistent contrails are formed within pre-existing cirrus. This is a complex topic, hardly researched at all, and with an ambiguous outcome. So, my understanding is that for this category, we do not have a clear ‘direction of travel’. That is OK, it should be more clearly acknowledged and a recommendation made as how to address this with future research.

I think the definition/explanation of “potential climate impact” is better placed here, not in the discussion, if such categorization is to be placed in the results. In Table 2, the entry “no” is confusing. I don’t know what this means. The authors omit to tell the reader that “potential climate impact” is a

loose approximation. There will be times that, for example, solar zenith angle will reverse findings via radiative transfer calculations. In (related) L195-L196 for the optically thick cirrus cloud, a “minor effect” is cited. This is vague. If the potential outcome is ambiguous, please say so. It could be cooling or minor warming.

Reply: As indicate in the introduction to this reply-to-reviewers, we placed the concept of climate impact at the beginning of the results section. Thanks to the input from Reviewer #3, we could add references to the expected increase of radiative forcing by contrails placed in clear sky compared to those added to existing cirrus (Spangenberg et al. 2013). As described above, categorises ‘stronger climate impact’ and ‘weaker climate impact’ are replaced by the categories ‘warming’ and ‘ambiguous’.

L149-150 States: “That means, most contrail-cirrus potentially form in regions already covered by natural cirrus”. Again, it is unclear what the outcome of this will be. If the pre-existing cirrus is ‘optically thick’, additional contrail cirrus MAY further thicken an already optically thick (naturally cooling) cloud further. But this is not clear unless more is known, and a full RT calculation is undertaken. All this is incredibly important as this could turn ‘conventional wisdom’ on its head about the net warming effect of contrail cirrus, or at the least, provide observational indirect evidence for a much smaller net warming (since most radiative transfer calculations are made under clear sky conditions). But we simply do not know from these observations. There is some (neglected here) further suggestive, if not definitive, evidence from the modelling study of Verma and Burkhardt (ref 39) who state: “Since corrections in ice nucleation are large in clouds with large ice water content, which are likely connected to large optical depth, *we speculate that disregarding the impact of cirrus on contrail formation may possibly underestimate the cooling of the contrail perturbations within optically thicker clouds*. It should be noted that the cirrus optical depth at which a further increase in optical depth, caused by contrail formation, leads to a cooling is dependent on the cirrus ice crystal habit.” (in ref 39’s conclusions). [* to * is this reviewer’s emphasis]

L193 “change” should this be “increase”?

Reply: Yes, indeed; we changed accordingly.

L200 “relevance is rated as limited”...my understanding is that Verma and Burkhardt (ref 39) found that the additional water mass of ice crystals had a negligible effect (in agreement with earlier work of Gierens (2012, see reference above). If I am correct, would it be useful to be clearer as the sentence addresses ‘climate impact’ through to ‘vortex phase’ with little explanation.

L204 “significant effect” please be more explicit.

Reply to both remarks: We clarify the explanation and modified the entire starting paragraph of the Discussions to (from line 280 to 288):

“The effects of contrails interacting with pre-existing cirrus clouds are generally known but have not yet been considered in the quantification of the contrail-cirrus climate impact. To start with, model studies reported that pre-existing cirrus have no impact on the formation of contrails, that contrail formation within cirrus mostly leads to increasing cirrus ice crystal numbers, that cirrus ice crystals mixed into the evolving contrail do not efficiently slow down the sublimation of the ice crystals in the downward travelling wing vortex, and that cirrus can dissolve an embedded contrail after few hours by aggregation of ice crystals. The changes in the radiative properties of subvisible and visible cirrus resulting from the above-described changes of cloud microphysics after the release of a contrail of mean optical depth of 0.34 (median 0.24, modal 0.1) into the cloud will increase the resulting optical depth of the modified cirrus (see Figure 1), but with yet unknown radiative impact. “

L219-225 Figure 4. Please be clearer over what you mean by “stronger” and “weaker” impact. By “stronger”, I am interpreting a greater potential climate impact (as you define it), which I interpret as a positive RF. I am less clear over “weaker” – weaker than what? Is it a cooling impact? Or a less warming impact than clear sky conditions/subvisible conditions? As per above, it seems as if adding optical thickness to an already optically thick cloud (which is ‘cooling’) as the negative forcing of the natural cloud, is made ‘more negative’ which then becomes a negative RF effect (as per the definition of RF, i.e. something interfering with the pre-industrial radiation balance).

Reply: Former Figure 4 was completely revised according to the criticism by all three reviewers. The now Figure 1 and accompanying text is now put to the front of the results section. Concerning the categories ‘stronger impact’ and ‘weaker impact’ we refer to the general text at the beginning of this document.

L227-232. Same comment over “stronger” and “weaker”. I’m not sure how much value this figure is adding, and if short of space, might be relegated to the SI.

Reply: See response above and introductory text. The figure itself was simplified and modified, following also a recommendation of Reviewer 1.

L234. Interesting statement. I don’t disagree from the analysis presented, but this offers quite a different perspective of the ‘big hits’ terminology (of which I remain unconvinced).

Reply: The paragraph containing this statement has been removed, see also our replies to the next points.

L235 I disagree. I think that suggesting this is premature. What is more to the point is that all this needs to be better understood before formulating strategies (which may turn out to be pointless). L236 says this better.

Reply: As said above, this statement has been removed. However, we kept the recommendation to include cirrus clouds in rerouting strategies, see also our final remark below.

L238 “should help minimizing trade-off effects...” I think this is subtly wrong. I think accounting for all this reduces risks of doing the wrong thing. At the moment, given the authors’ results, the outcome of re-routing has just become far more uncertain.

Reply: As already stated, this sentence has been removed.

L234-248. This all makes sense, but it would be nice to have a stronger recommendation as to what research would firm this up (ERF vs occurrence conditions). Can this be accounted for/studied in global models or can some hybrid observational/radiative transfer study be undertaken?

Reply: Agreed, see our response below.

L242-248. This seems like people-pleasing. We do not know - and this paper adds to the evidence – that re-routing is either a sensible or viable option for mitigating aviation’s impacts on climate. As they say, “get off the fence”! More seriously, simply reiterate the importance of these findings for re-routing proposals.

Reply: Very good suggestion! We removed the entire paragraph on the rerouting and kept only the final statement (lines 325 – 328): “This topic needs to be further analysed and verified in terms of their radiative impact for consideration of the potential impact of contrail avoidance strategies including flight rerouting. Finally, ice-supersaturation only is by far not a reliable, stand-alone criterion for

assessing contrail-cirrus climate effects. Instead, we need to include pre-existing cirrus clouds and their interactions with contrail-cirrus into all our considerations.”

IN CONCLUSION

I think that the findings are of great importance and significance, but I am uncomfortable with their simplistic treatment of “potential climate impact”. This seems relatively clear, based on prior work with RT models/GCM calculations with RT, that clear sky conditions will net warm. Similarly, this may be the case for sub-visible cirrus with low OD. For persistent contrails forming contrail cirrus clouds within pre-existing cirrus, which seems to be the majority of incidence of persistent contrails from this study, then without much more information, we don’t know what the outcome of this will be. This should be stated much more clearly, and provide a recommendation for further work, given that the authors’ findings suggest that we should have an increased concern over what was already known to be a huge uncertainty – formation of persistent contrails within pre-existing cirrus. The title of the article says it all!

Reply to all comments on the concluding section (lines 234 to 248 and Conclusions):

The entire concluding section was rewritten, taking up the points raised by all reviewers. Please refer to the revised manuscript, starting on line 317.

Reviewer #3 (Remarks to the Author):

Summary of the manuscript:

“Contrails inside cirrus clouds predominate with uncertain climate impact” by Petzold et al. examines the potential occurrence of contrails, short-lived or persistent, as a function of humidity and ambient cirrus occurrence. Contrails form when the Schmidt-Appleton criterion is met and nominally can persist in ice-supersaturated regions (ISSR) where $RH_{ice} > 100\%$. Recent research shows that contrails can also persist when $RH_{ice} > 90\%$, giving rise to a larger range of humidities and, hence, a greater number of regions (43% more) of potential contrail coverage. Those regions are designated as potential contrail-cirrus regions (PCCRs). Results are presented for both ISSRs and PCCRs.

The motivation for the study is that a certain class of tropospheric conditions could be eliminated from any aircraft rerouting system that might be established to minimize contrail effects. That particular class of conditions is defined as the presence of thick cirrus clouds at planned flight levels. For this class, the radiative forcing induced by the development of contrail cirrus clouds is assumed to be small, or negligible enough to be ignored. Ergo, there is no need to reroute the flight because of contrail effects.

The analysis uses 7 years of IAGOS aircraft RH_{ice} measurements matched with interpolated ERA 5 to determine the ambient cloud conditions at flight level. A threshold of ERA5 CIWC > 0.001 ppmv is used to differentiate between clear and cloudy conditions. To further classify the cloud conditions, a cloud optical depth (OD) of 0.03, equivalent to CIWC = 1.0 ppmv, was used as the threshold differentiating sub-visible from thick cirrus for a given cloudy datapoint. Thus, any preexisting cirrus with cloud OD > 0.03 is assumed to be thick cirrus. Radiative forcing (RF) as a function of cloud OD is characterized using a plot of RF as a function of cirrus OD provided by Ref 11. The plot reproduced from Ref 11 indicates that RF increases with cloud OD up to a value of ~ 0.85 then drops precipitously for greater cirrus ODs to negative values for OD > 1.0 .

The results of the analysis show that for the 4 regions examined, only 20% (ISSRs) - 30% (PCCRs) of the air masses can support contrail cirrus formation. Of these, roughly 64% of the contrails have a weaker impact on RF (climate) and the remaining 36% have a stronger impact. The overall effects vary regionally and seasonally, with minima over SE Asia and maximum effects during spring and winter. The results lead to the recommendation that any plan to adjust flight levels should include model predictions of cirrus locations, so that planes could fly in areas with thick cirrus and height adjustments would only be needed for clear areas and those with sub visible cirrus. The results also suggest that $\sim 90\%$ of contrails form within thick cirrus over SE Asia, it would probably be of little benefit to undertake any flight rerouting.

Review:

The uncertainty in radiative forcing by contrails is high, leading to low confidence in our understanding of this somewhat small component of the climate change equation. Rerouting aircraft to avoid forming contrail cirrus is probably the most straightforward solution to minimizing contrail effects. However, it involves expenditure of more fuel and would further complicate air traffic control procedures. Thus, minimizing the number of possible aircraft reroutes by eliminating suspect air masses is a worthy effort. Thus, the goal of this study is of interest to the aviation and climate communities.

The premise is logical: contrails forming in a clear sky or in sub visible cirrus should have greater radiative impacts than those forming in thick cirrus. Persistent contrail development in already thick

cirrus adds little to either the LW or SW impact of the existing cirrus because of the small change in OD for optically thick cirrus clouds. Yet, there are some shortcomings in how this premise is addressed.

1) The first issue is that the authors looked to no later than 2010 (ref 29) for information about contrail cloud overlap. There are observations of persistent contrail coverage available for 2006 and 2012 (Duda et al., 2013; 2019).

Duda et al. (2013) combined contrail analyses of 1 year of Northern Hemisphere Aqua MODIS data with MODIS cloud retrievals (Minnis et al. (2011)). Using these, Bedka et al. (2013) differentiated between contrails over/within ice clouds (cirrus) and contrails detected in clear skies or over low-level water clouds. Passive satellite retrievals have difficulty detecting cirrus with $OD < 0.05$. The hit rates for clouds having $OD < 0.1$ are around 50% (Trepte et al., 2019). Thus, the clear category would include both clear and subvisible clouds and the ice clouds would be thick according to the definition used here. With that in mind, Bedka et al. (2013) found that contrails were detected with ice clouds predominating in 33% of the cases, compared to 46% with low liquid water clouds, and 17% in clear skies. Seasonal variations are superimposed on those averages. There are likely a few more percent contrails having ice cloud concurrence as $\sim 20\%$ of clouds classified as water were likely to consist of non-opaque cirrus over lower liquid water clouds and a few more water clouds were misclassified as ice than ice clouds misclassified as water (Sun-Mack et al., 2024). Nevertheless, the observed fraction of weaker impact conditions (33% coincidence with ice clouds) is significantly smaller than the 64% computed here. Perhaps, the selection of $CIWC = 0.001$ ppmv as the cloud threshold produces too many clouds. It is not all that different from 0.000 ppmv in frequency. This would be consistent with Reference 57, which indicates that ERA5 generates too much ice cloud in the upper troposphere relative to CALIPSO.

Reply: The selected CIWC threshold of 0.001 ppmv is based on a large set of in-situ observations of total water and gas-phase water from in total 24 field campaigns and validated by analysing the RH_{ice} distributions from IAGOS for various CIWC thresholds; see 'Methods' section. Since the selected threshold is based on in-situ observations, we believe it is justified. Furthermore, we combine clear sky ($CIWC < 0.001$ ppmv) and subvisible cirrus ($0.001 \text{ ppmv} \leq CIWC < 1.0 \text{ ppmv}$) to one category when discussing potential climate impacts. In that respect, the details of the selected threshold have no impact on our conclusions.

More influential on the results seems to be the visibility threshold and the selected area. On request by Reviewer 1 we combined the former regions 1 – 3 into one single region for Northern Midlatitudes and recalculated our analysis. Now we find a contrail-cirrus overlap of 55% to 60% for a CIWC visibility threshold of 1.0 ppmv and 46% to 52% for a threshold of 2 ppmv. It looks like the cloud visibility threshold of 2 ppmv developed from Sassen et al. (1992) is closer to the T_{11} criteria used for the analysis by Bedka et al..

We discuss the results for contrail - cirrus overlap; see line 247 – 254: "The above-mentioned study on properties of linear contrails using MODIS data reports 33% of contrails formed in areas already covered by clouds. The applied method distinguishes contrails from other cirrus by their linear shape and relatively large 11–12 μm brightness temperature difference. We apply a combination of in-situ based criteria such as the fulfilment of SAC, exceedance of RH_{ice} threshold for PCCR, and exceedance of ERA5 $CIWC_{vis}$ to an on-flight-level dataset and obtain a fraction of 46% (PCCR, $CIWC_{vis} = 2.0$ ppmv). Taking into account that about $\sim 20\%$ of clouds classified in the MODIS analysis as water were likely to consist of non-opaque cirrus over lower liquid water clouds and a few more water clouds were misclassified as ice than ice clouds misclassified as water, the agreement between the two studies is reasonable."

Additionally, Table 1 lists the results for both visibility thresholds to indicate the uncertainty in the obtained values.

Spangenberg et al. (2013; S13) computed RF for all of the contrails. The resulting mean values support the premise of this paper. Indeed, one could look at those averages and come to the same conclusion as the results of this paper: Flying in an atmosphere with preexisting visible cirrus will cause less RF than flying in other conditions. In fact, the observations actually quantify the differences. S13 found that the net unit contrail RF during the daytime (~13:30 LT) was 3.9 Wm⁻² for clear skies and 1.3 Wm⁻² for contrails in the same column as ambient cirrus clouds. That is a factor of 3 decrease, a change that would support the use of the sub visible threshold. Yet, the LW RFs for clear and cloudy scenes are 18.5 and 11.7 Wm⁻², respectively, a difference of only 36%. That LW difference is the same at night when no SW RF occurs to balance the LW RF.

Reply: Thank you very much for pointing us to that highly valuable reference. We added the following paragraph on lines 46 to 54: “The analysis of linear contrail occurrence over the Northern hemisphere in MODIS data for the year 2006 revealed, that contrails were detected within an environment with ice clouds in 33% of the cases, compared to 46% with low liquid water clouds, and 17% in clear skies. The same data set yields that the net contrail radiative forcing (RF), i.e., the sum of short-wave and long-wave RF, during daytime (~13:30 LT) is 3.9 Wm⁻² for clear sky and 1.3 Wm⁻² for contrails in a column already filled with visible cirrus clouds. During night, when no shortwave RF occurs, the longwave RF for clear and cloudy scenes is 16.9 and 11.0 Wm⁻², respectively. Thus, daytime net contrail RF is increased by a factor of 3 while nighttime longwave RF still is increased by a factor of 1.5 for contrails formed in clear sky or subvisible cirrus, compared to those having formed inside, above or below pre-existing cirrus.”

These findings were used as justification for our categorisation as “warming” and “ambiguous” in the description of our methodology, described from line 134 to line 159. We refer the reviewer to the manuscript because this section is too long to be included here. The justification itself can be found on lines 156 – 159: “The net warming of Cases 1 and 2 and the minor net warming of Case 3 are confirmed by MODIS observations, whereas a model study speculates that ignoring the impact of cirrus on contrail formation may possibly underestimate the cooling of the contrail perturbations within optically thicker clouds (Case 4).”

2) The premise of this paper seems too limited in scope. One would expect that a contrail developing above or below a thick cirrus would also have a similar impact as a contrail imbedded in the cirrus since the ice cloud OD added to the total column is nearly the same in both cases. Perhaps, that is implied here somehow.

Reply: The focus on our study is on the fraction of contrail-cirrus forming inside pre-existing clouds, determined from in-situ observations of relative humidity with respect to ice. Given the nature of in-situ sampling by aircraft, we can only collect information along the flight path but have no access to information about cloudiness etc. above or below flight altitude. The reviewer states correctly that a contrail developing above or below a pre-existing cloud also has a different radiative impact than a contrail forming in a clear sky column, we cannot investigate these effects in our study.

We added the following paragraph at the beginning of the main text to clarify the scope our study which starts on line 37: “Uncertainties of the climate effects of contrail-cirrus are largely related to the unanswered question of the extent to which contrails form inside cirrus clouds, how they influence cirrus cloud coverage and optical depth, and what are the radiative implications. However, it is obvious that contrail-cirrus formed in clear sky or in or above pre-existing subvisible cirrus clouds have a stronger

radiative effect than contrail-cirrus embedded in pre-existing visible cirrus, because the difference in the optical contrast with and without contrail is much larger.”

We finish the Method section with a

“Final note on data interpretation

Given the nature of in-situ sampling by aircraft, we can only collect information along the flight path but have no access to information about cloudiness etc. above or below flight altitude. Knowing that a contrail developing above or below a pre-existing cloud also has a different radiative impact than a contrail forming in a clear sky column, we cannot investigate these effects in our study. Thus, the reported fractions of contrail-cirrus inside clouds should be taken as a lower limit of the contrail-cirrus interacting with pre-existing cloudiness.”

3) The somewhat vague notion of weaker and stronger impacts is not particularly satisfying. Given the observed RF results, what average difference in RF is considered to be enough to separate weak and strong impacts? Is the use of cloud OD > 0.03 as a thick cirrus optimal? Is the characterization “thick” warranted or should visible be used always. Normally one would consider a cirrus cloud to be thick when the emissivity approaches unity or the change in emissivity with unit optical depth is relatively small, typically when OD is between 3 and 6. While it is true that adding a contrail to sub visible cirrus will have nearly the same impact as adding a contrail to a clear-sky, there could still be significant contrail RF impacts when the extant cirrus has an OD < 1.5 (e.g., Meerkötter et al.1999). Hong and Liu (2015) show that a large portion of ice clouds observed by CALIPSO have ODs between 0.05 and 1.5, while the mode of the total distribution is OD = 1.2. Thus, there will be a range of significant RFs that could be produced by contrails. Determining the optimal cirrus OD threshold would require a sensitivity study and possibly consideration of low-level clouds.

This need for quantification is more apparent when considering that the weaker impacts constitute almost 100% of persistent contrail diagnoses in SE Asia. That suggests that overall RF should be smaller in that region compared to North America or Europe. The RF results from S13 suggest otherwise. They suggest that over the SE Asia flight corridors are much like their USA and European counterparts, and even greater at night.

Reply: The main outcome of our study is the quantification of the clear-sky to subvisible cirrus areal fraction close to ice-saturation which allow for the existence of long-living contrail-cirrus. However, given our data set, we cannot quantify the RF threshold between weak and strong climate impact. Instead, we moved to warming and ambiguous climate impacts as qualitative descriptions. Nevertheless, the radiative impacts need to be studied. Thanks to the clear comments by all reviewers, we put this statement prominently into our conclusions (lines 317 to 328) since this is one of the major outcomes of our study:

“The results of our study suggest that conditions necessary for the formation of contrail-cirrus are most often fulfilled in regions that are already covered by visible cirrus, which reduces their warming impact or even turns it into a cooling effect. Those of a stronger warming significance, however, form under only limited circumstances in clear sky and subvisible cirrus. Unfortunately, our knowledge on the radiative effects of contrail-cirrus embedded in subvisible, visible and optically thick clouds, and feedback mechanisms between contrail and cirrus ice particles is highly limited. There will be a range of radiative forcings that could be produced by embedded contrail-cirrus, and the quantification of the climate impacts would require a sensitivity study and possibly consideration of low-level clouds.

This topic needs to be further analysed and verified in terms of their radiative impact for consideration of the potential impact of contrail avoidance strategies including flight rerouting. Finally, ice-supersaturation only is by far not a reliable, stand-alone criterion for assessing contrail-cirrus climate effects. Instead, we need to include pre-existing cirrus clouds and their interactions with contrail-cirrus into all our considerations.”

Other comments

It should be noted that the RF results in Fig. 4 are based on a single set of conditions, noon at SZA = 50 deg?. There is no consideration for the ERF at night, which is all warming. The results for other angular conditions, latitudes, and backgrounds will be highly variable.

Reply: This information has been added to the text connected to the description of Figure 4. We added the following sentence (lines 145 to 148): “Since only a single set of conditions, (noon at solar zenith angle at 50° North) was used and the results for other angular conditions, latitudes, and backgrounds will be highly variable, the figure only serves as a descriptive sketch. The radiative effects of contrail-cirrus developing in the various environments are categorised in the context of radiative effects of the embedded contrail-cirrus in addition to the radiative effects of the unperturbed clouds.”

Line 331 red curve is for 0.01, not 0.05. Purple curve is for 0.05 in Fig. &a.

Reply: Thank you, corrected.

Despite the issues and reservations described above, not to mention the uncertainties in current model representations of RH_{ice} and clouds, the authors have laid the groundwork for a reasonable method to define cirrus cloud conditions that could theoretically be used to reduce the magnitude of flight rerouting in a future contrail avoidance system.

References cited in the response to the Reviewers

Bedka, S. T., Minnis, P., Duda, D. P., Chee, T. L., and Palikonda, R.: Properties of linear contrails in the Northern Hemisphere derived from 2006 Aqua MODIS observations, *Geophysical Research Letters*, 40, 772-777, 10.1029/2012gl054363, 2013.

Bickel, M., Ponater, M., Burkhardt, U., Righi, M., Hendricks, J., and Jöckel, P.: Contrail Cirrus Climate Impact: From Radiative Forcing to Surface Temperature Change, *Journal of Climate*, 38, 1895-1912, 10.1175/jcli-d-24-0245.1, 2025.

Prather, M. J., Gettelman, A., and Penner, J. E.: Trade-offs in aviation impacts on climate favour non-CO₂ mitigation, *Nature*, 8, 10.1038/s41586-025-09198-2, 2025.

Singh, D. K., Sanyal, S., and Wuebbles, D. J.: Understanding the role of contrails and contrail cirrus in climate change: a global perspective, *Atmospheric Chemistry and Physics*, 24, 9219-9262, 10.5194/acp-24-9219-2024, 2024.

Spangenberg, D. A., Minnis, P., Bedka, S. T., Palikonda, R., Duda, D. P., and Rose, F. G.: Contrail radiative forcing over the Northern Hemisphere from 2006 Aqua MODIS data, *Geophysical Research Letters*, 40, 595-600, 10.1002/grl.50168, 2013.

Verma, P. and Burkhardt, U.: Contrail formation within cirrus: ICON-LEM simulations of the impact of cirrus cloud properties on contrail formation, *Atmospheric Chemistry and Physics*, 22, 8819-8842, 10.5194/acp-22-8819-2022, 2022.

Zhang, W. Y., Van Weverberg, K., Morcrette, C. J., Feng, W. H., Furtado, K., Field, P. R., Chen, C. C., Gettelman, A., Forster, P. M., Marsh, D. R., and Rap, A.: Impact of host climate model on contrail cirrus effective radiative forcing estimates, *Atmospheric Chemistry and Physics*, 25, 473-489, 10.5194/acp-25-473-2025, 2025.

RESPONSE TO REVIEWERS' COMMENTS

Responses to final reviewer's remarks are marked in green. The annotated manuscript is added to this document.

Reviewer #1 (Remarks to the Author):

The authors have carefully revised their manuscript according to the comments provided by three reviewers. The paper is now very straightforward to read and conveys a clear message. The use of Figure 1 for illustrating the conceptual expectation is good.

Going through the manuscript, I found it difficult to reconcile the information provided in Figures 3 and 5, which are clearly showing different combinations of the same numbers. While all became clear after inspecting Table 1, I would try to avoid similar confusion with readers by omitting Figure 5 from the paper.

Overall, this is great and important work that sheds some light on an understudied phenomenon. I am happy to see this published.

Reply: Thank you very much for the very positive feedback, we very much appreciate that. The idea behind showing these two different figures is that

- 1. in Figure 3, we focus on the separation of no contrail formation, short-lived contrails, and long-lived contrails, distinguished in each case according to clear-sky and in-cloud conditions,*
- 2. in Figure 5, we focus on the climate effect according to our conceptual analysis and split up in areas of no impact, ambiguous impact and warming impact.*

Since we still believe in the visual strength of Figure 5, we want to keep it. To avoid confusion, we clarified the different purposes of Figs. 3 and 5 by rephrasing the sentence describing Fig.5, starting on line 282. It reads now:

“The results summarised in Figure 5 and presented numerically in Table 1, middle column, highlight the expected climate impact. This information supplements that given in Figure 3 which focuses on the environmental conditions of contrail-cirrus occurrence. ...”

Reviewer #2 (Remarks to the Author):

The authors have responded comprehensively and thoroughly to all three reviews and I recommend publication as soon as possible of the ground-breaking paper.

I would only suggest one tiny modification, where the authors have (I think) possibly misunderstood my point. My comment was "L30 please insert "present day" after 60%, since the proportion depends entirely on recent growth rates of aviation (there is a misconception that this is somehow a fixed proportion)."

The authors replied: We changed the sentence to "With an ERF of 60 mW m⁻² contrail-cirrus is estimated to be the largest non-CO₂ ERF contributing 60% (present-day knowledge) to the aviation net total ERF of 100 mW m⁻²".

I was not referring to "present-day knowledge" (and proportions, thereof) of aviation ERF assessments but rather that the proportion of non-CO₂ forcing to the total is a function of the history of CO₂ emissions and the complexities of C cycle CO₂ accumulation (and subsequent ERF) and the simpler linear response of SLCFs of ERF to emissions (in general). This can be seen in Figure 2c of Klower et al. (2021) <https://doi.org/10.1088/1748-9326/ac286e> (RH bars), where different growth rate scenarios result in different proportions of response (it's actually dT but that matters little to the point being made).

The authors do not need to add a reference or anything like that, I only point out to clarify and they can very simply fix this by removing the word "knowledge".

Otherwise, great job!

Reply: Thank you very much for the highly encouraging feedback. The modification was implemented as suggested by removing the word "knowledge" on line 30.

Reviewer #3 (Remarks to the Author):

The authors have satisfactorily answered my concerns and altered the paper accordingly. Thus, would recommend publication in its current form.

Reply: Thank you very much for the very positive feedback.